# Route to chaos and resonant triads interaction in a truncated rotating nonlinear shallow–water model

**Francesco Carbone**[1]☺, **Denys Dutykh**[2,3]☺ *

**1** National Research Council - Institute of Atmospheric Pollution Research, C/o University of Calabria, Rende, Italy, **2** Mathematics Department, Khalifa University of Science and Technology, Abu Dhabi, United Arab Emirates, **3** Causal Dynamics Pty Ltd, Perth, Australia

☺ These authors contributed equally to this work.
\* denys.dutykh@ku.ac.ae

**Data Availability Statement:** https://zenodo.org/records/11367257.

**Funding:** FSU-2023-014.

## Abstract

The route to chaos and the phase dynamics of the large scales in a rotating shallow-water model have been rigorously examined through the construction of an autonomous five-mode Galerkin truncated system employing complex variables, useful in investigating how large/meso-scales are destabilized and how their dynamics evolves and transits to chaos. This investigation revealed two distinct transitions into chaotic behaviour as the level of energy introduced into the system was incrementally increased. The initial transition manifests through a succession of bifurcations that adhere to the established Feigenbaum sequence. Conversely, the subsequent transition, which emerges at elevated levels of injected energy, is marked by a pronounced shift from quasi-periodic states to chaotic regimes. The genesis of the first chaotic state is predominantly attributed to the preeminence of inertial forces in governing nonlinear interactions. The second chaotic state, however, arises from the augmented significance of free surface elevation in the dynamical process. A novel reformulation of the system, employing phase and amplitude representations for each truncated variable, elucidated that the phase components present a temporal piece-wise locking behaviour by maintaining a constant value for a protracted interval, preceding an abrupt transition characterised by a simple rotation of $\pm\pi$, even as the amplitudes display chaotic behaviour. It was observed that the duration of phase stability diminishes with an increase in injected energy, culminating in the onset of chaos within the phase components at high energy levels. This phenomenon is attributed to the nonlinear term of the equations, wherein the phase components are introduced through linear combinations of triads encompassing disparate modes. When the locking durations vary across modes, the resultant dynamics is a stochastic interplay of multiple $\pi$ phase shifts, generating a stochastic dynamic within the coupled phase triads, observable even at minimal energy injections.

**Competing interests:** The authors have declared that no competing interests exist.

# 1 Introduction

Fluid turbulence on the geostrophic scale is characterized by the simultaneous presence of turbulent and wavelike motions, which both depend on the effects of planetary rotation [1, 2]. Moreover, their interplay is related to essential processes in atmospheric and oceanic sciences, such as transport and exchange of moisture, heat, gaseous tracers or salinity, and momentum, depending on the which of boundary layer being considered [3–5] (*i.e.* oceanic or atmospheric). Such kind of flows may exhibit multiple forms of motion over a wide range of length and time scale [3, 6, 7], and their global dynamics is the result of several entangled phenomena of different nature, including the formation of extreme events such as cyclones and tsunamis.

Generally, the vertical fluid accelerations of these boundary layers are considered negligible, while horizontal motion is considered to be in hydrostatic balance. So, within this framework, the rotating shallow water (RSW) model is seen as a classical modelling tool for investigating a plethora of dynamic phenomena taking place in the atmosphere or oceans. The RSW, consisting of a system of hyperbolic/parabolic partial differential equations which describes the spatiotemporal evolution of a flow below a pressure surface in the fluid (not necessarily restricted to a free surface), represents a simplified prototype model able to capture the complexity of the fully three-dimensional rotating flow. Indeed, such a family of models represents a physical simplification obtained from depth-integrating the full equations of the stratified rotating fluid, the so-called "primitive equations" [2], so that the flow is constrained on a two-dimensional geometry. The RSW system can show complex emergent dynamics due to quadratic nonlinearities, such as wave mixing, local/non-local turbulent energy transfer between disparate scales, formation of intermittent events or extreme bursts, and small-scale dynamics (shocks or wave-braking).

Nevertheless, few studies are focused on understanding how the inverse energy flow (directed toward large scales), or resonant interactions, can act as a destabilizing effect for large and mesoscale flow, thus producing strong amplitude oscillations in the so-called "coherent structures" [8], which may finally drive the system through chaotic dynamical states. In fact, a necessary condition for observing a chaotic behaviour in physical dynamical systems is to have one nonlinear term [9], and at least three physical variables (*e.g.* the classical low-order Lorenz systems [10, 11]).

Despite the complexity of such emergent dynamics, the large-scale behaviour can be well understood and captured by simple dynamical models, far from the inertial ranges of fully developed turbulence. Generally, such models are constructed by following the classical Galerkin dimension reduction, obtained through a truncation to a finite (low) number of modes in the Fourier space. Truncated Galerkin models of incompressible hydrodynamic equations have been widely investigated since the pioneering work of Ruelle and Takens [12], where in the framework of a "dynamical systems approach" to turbulence [13], the way chaotic orbits settle down to a chaotic attractor, in a turbulent flow described by the Navier-Stokes equations, have been investigated, leading to the formalization of the concept of "strange attractors" [14].

While intriguing, the Galerkin truncation methodology is not without its challenges. Specifically, employing an insufficient number of modes could lead to a distortion in the observed phenomenology. Consequently, efforts to ascertain the minimal set of modes necessary for stabilizing the dynamics by adding or substituting modes have yielded uncertain outcomes, often suggesting a number significantly beyond what might be necessary [15, 16]. This is primarily attributed to the non-universality of Galerkin approximations [17–19]. It should be remarked that Galerkin models do not contain, nor shall we be concerned with, the inertial or dissipative ranges of fully developed turbulence, whose description requires an infinite number of wave vectors. Nonetheless, such models have successfully elucidated certain experimental

phenomena [20–26]. These models offer valuable insights into the macroscopic behaviour characteristic of the onset of turbulence [27–29], encompassing both finite and/or infinite sequences of bifurcations [30], as well as delineating a quasi-periodic pathway to low-dimensional chaos in dissipation-dominated dynamical systems [31–33], and to intermittent chaos in nearly conservative systems [34].

In the framework of the RSW, the apparent limitation of the truncated model lies in the inability to study the formation of small scales. As well known, direct numerical simulations show that shocks strongly contaminate turbulence in RSW, and predictions of the wave-turbulence theory are thus difficult to verify. In fact, direct low-order Galerkin truncations could not capture the wave-breaking, making any distinction between wave and vortex modes, which, however, is far from the objectives of this work; nonetheless, despite the limitations, truncated models are still useful in investigating how large/meso-scales are destabilized and how their dynamics evolves and transits to chaos.

In this context, the route to chaos within RSW flow has been meticulously explored through the implementation of a complex five-mode Galerkin approximation. This analysis reveals that the chaotic transition in the system unfolds in a manner distinctively intriguing when compared to its fluidic counterparts. Special emphasis has been placed on examining the dynamics of phase and amplitude pertaining to the various constituents of the RSW system. It has been demonstrated that the essence of chaos is encapsulated in the simultaneous presence of nonlinear interdependence, determinism, and an inherent order guiding the systems' trajectory evolution. Moreover, a perpetual oscillation exists between regions exhibiting step-like behaviours and continuous phase-locking alongside random phase switches within the system's phase variables.

## 2 Coupling triads approximation for the rotating shallow water model

Let us consider a layer of fluid in a two-dimensional rotating frame of reference, subject to the gravitational field, whose dynamics can be described through the forced-dissipative RSW:

$$\frac{\partial \mathbf{u}}{\partial t} + (\mathbf{u} \cdot \nabla)\mathbf{u} = -g\nabla\eta + \mathbf{\Omega} + \mathcal{D} + \mathcal{F} \tag{1}$$

$$\frac{\partial \eta}{\partial t} + (\mathbf{u} \cdot \nabla)\eta + \eta(\nabla \cdot \mathbf{u}) = -H(\nabla \cdot \mathbf{u}) \tag{2}$$

where $\mathbf{u}(\mathbf{r}, t) \equiv \{u(\mathbf{r}, t), v(\mathbf{r}, t)\}$ is the depth-averaged fluid velocity field, $\eta(\mathbf{r}, t)$ is the displacement from the reference height $H > 0$, and $g$ is the gravity acceleration. The system evolves on a two-dimensional torus (*i.e.* periodic boundary condition are imposed), and the operator $\mathcal{D}$ represents the viscous dissipative term, which for simplicity has been defined as $\mathcal{D} \equiv \nu\nabla^2\mathbf{u}(\mathbf{r}, t)$ ($\nu$ is the kinematic eddy viscosity), $\mathbf{\Omega} \equiv \{fv(\mathbf{r}, t), -fu(\mathbf{r}, t)\}$, represents the rotation term ($f$ is Coriolis frequency), and $\mathcal{F} \equiv \{F_u, F_v\}$ represents an eventual external forcing term acting in principle on both $u$ and $v$. The square modulus $|\mathcal{F}|^2$ represents the amount of energy injected into the system per unit time and mass.

In the absence of external forcing and dissipative term, since the RSW forms a (non-canonical) Hamiltonian system, an exact energy balance can be derived for the system of Eqs (1) and (2). Indeed, it can be shown that the domain averaged total energy density $\mathcal{E}(t)$, defined as:

$$\mathcal{E}(t) = \frac{1}{2L^2} \int \{h(\mathbf{r}, t)[u^2(\mathbf{r}, t) + v^2(\mathbf{r}, t)] + gh^2(\mathbf{r}, t)\} \, d\mathbf{r} \tag{3}$$

is a constant of the motion [35], where $h(\mathbf{r}, t) = H + \eta(\mathbf{r}, t)$. As a further conserved quantity, by taking the curl of the momentum Eq (1) in combination with the continuity Eq (2), the classical relation for the conservation of the potential vorticity $\mathcal{Q}(t)$ can be obtained [35]:

$$\mathcal{Q}(t) \equiv \frac{D}{Dt}\left(\frac{\zeta(\mathbf{r}, t) + f}{h(\mathbf{r}, t)}\right) = 0, \tag{4}$$

where $\zeta(\mathbf{r}, t) \coloneqq \partial_x v(\mathbf{r}, t) - \partial_y u(\mathbf{r}, t)$ represents the relative vorticity, and $D/Dt \coloneqq \partial/\partial t + (\mathbf{u} \cdot \nabla)$ is the well known material derivative. In the simplest case, unforced and inviscid, the linearized RSW equations support the presence of plane waves propagating with characteristic phase speed $c_s = \sqrt{Hg}$. Moreover, three characteristic times can be recognized, say $f^{-1}$ related to the Coriolis parameter, the dissipative time $t_D = H^2/\nu$ and the characteristic period (linear time) of the gravity waves propagation $\tau_w = \sqrt{Hg^{-1}}$.

In the wave vector space, in terms of Fourier representation, the fields $\mathbf{u}(\mathbf{r}, t)$ and $\eta(\mathbf{r}, t)$ can be expressed in terms of Fourier coefficients $\{\mathbf{u}_k(t), \eta_k(t)\}$ as:

$$\mathbf{u}(\mathbf{r}, t) = \sum_{k=1}^{\infty} \mathbf{u}_k(t)\exp[\mathrm{i}\mathbf{k} \cdot \mathbf{r}] + \text{c.c.}, \tag{5}$$

$$\eta(\mathbf{r}, t) = \sum_{k=1}^{\infty} \eta_k(t)\exp[\mathrm{i}\mathbf{k} \cdot \mathbf{r}] + \text{c.c.}, \tag{6}$$

where $\mathbf{k} \coloneqq k_0\mathbf{n} \equiv (2\pi L^{-1})\,\mathbf{n}$, being $L$ the rectangular domain length, $\mathbf{n} \equiv (n_x, n_y)$ a set pairs integers ($n_x, n_y \in \mathbb{N}$), and c.c. represents the complex conjugate operation. By substituting such definitions in Eqs (1) and (2) (projecting on the Fourier space), the system becomes an infinite set of ordinary differential equations for the complex amplitudes $\{\mathbf{u}_k(t), \eta_k(t)\} \in \mathbb{C}$, $\forall t \in \mathbb{R}^+$:

$$\frac{\mathrm{d}u_k}{\mathrm{d}t} + \mathrm{i}\sum_{\mathbf{p},\mathbf{q}}^{\Delta}\left[\mathbf{q} \cdot \mathbf{u}_\mathbf{p}\right]u_q = -igk_x\eta_k + fv_k - \\ -\;\; \nu k^2 u_k + F_u \tag{7}$$

$$\frac{\mathrm{d}v_k}{\mathrm{d}t} + \mathrm{i}\sum_{\mathbf{p},\mathbf{q}}^{\Delta}\left[\mathbf{q} \cdot \mathbf{u}_\mathbf{p}\right]v_q = -igk_y\eta_k - fu_k - \\ -\;\; \nu k^2 v_k + F_v \tag{8}$$

$$\frac{\mathrm{d}\eta_k}{\mathrm{d}t} + \mathrm{i}\sum_{\mathbf{p},\mathbf{q}}^{\Delta}\left[\mathbf{k} \cdot \mathbf{u}_\mathbf{p}\right]\eta_q = -\mathrm{i}H(\mathbf{k} \cdot \mathbf{u}_\mathbf{k}), \tag{9}$$

where $k = \sqrt{k_x^2 + k_y^2}$, and the sum in the nonlinear terms is a shorthand notation for:

$$\sum_{\mathbf{p},\mathbf{q}}^{\Delta} \equiv \sum_{\mathbf{p},\mathbf{q}} \delta_{\mathbf{k},\mathbf{p}+\mathbf{q}}.$$

Such summation is extended to all triads of wave vectors satisfying the triangular condition $\mathbf{k} = \mathbf{p} + \mathbf{q}$.

The resulting system of equations can be written in nondimensional form by normalizing lengths by $H$, times by $\tau_w$, and velocities by $c_s$, respectively, thus becoming:

$$\frac{\mathrm{d}u_k}{\mathrm{d}t} + \frac{\mathrm{i}}{2}\sum_{p,q}^{\Delta}[k_x u_p u_q \quad + \quad q_y v_p u_q + p_y u_p v_q] = -$$
$$-\mathrm{i}k_x\eta_k \quad + \quad \frac{v_k}{\mathrm{Ro}} - \frac{k^2}{\mathrm{Re}}u_k + \delta_{k,k_f}F_0 \tag{10}$$

$$\frac{\mathrm{d}v_k}{\mathrm{d}t} + \frac{\mathrm{i}}{2}\sum_{p,q}^{\Delta}[k_y v_p v_q \quad + \quad q_x u_p v_q + p_x u_q v_p] = -$$
$$-\mathrm{i}k_y\eta_k \quad - \quad \frac{u_k}{\mathrm{Ro}} - \frac{k^2}{\mathrm{Re}}v_k \tag{11}$$

$$\frac{\mathrm{d}\eta_k}{\mathrm{d}t} + \frac{\mathrm{i}}{2}\sum_{p,q}^{\Delta}[k_x(u_q\eta_p \quad + \quad u_p\eta_q) + k_y(v_q\eta_p + v_p\eta_q)] = -$$
$$-\mathrm{i}(k_x u_k \quad + \quad k_y v_k), \tag{12}$$

where $\mathrm{Ro} := (f\tau_w)^{-1} \equiv (f\sqrt{Hg^{-1}})^{-1}$ is the Rossby number, say the ratio between the rotation and the inertial times, and $\mathrm{Re} := Hc_s\nu^{-1}$ is the Reynolds number. The Kolmogorov length results to be $\ell_k := H\mathrm{Re}^{-3/4}$. For simplicity, the external forcing term was chosen to act only along the direction of the $u$ component of the velocity field and on a single wave vector $k_f$, and takes the form $F_0 \equiv F_u\,g^{-1} = A\exp[\mathrm{i}\pi/4] \in \mathbb{C}$.

The system of Eqs (10)–(12) contains only three free parameters, namely the Reynolds number Re, the Rossby number Ro and the amplitude of the external forcing $A$.

## 3 Low dimensional truncation

In the absence of forcing and dissipation, when the number of wave vectors involved in the nonlinear couplings is infinite, the two invariants of the system $\mathcal{Q}$ and $\mathcal{E}$ should survive for every single triad of interacting wave vectors or in every Galerkin truncation of the infinite System (10)–(12). If this condition holds, a finite truncated low–order model $\mathcal{T}_N(u,v,\eta)$, which maintains all global characteristics of the complete system, can be obtained by taking into account only a finite number $N$ of the interacting wave vectors $\mathbf{k}_i$ ($i = 1, 2, \ldots, N$), satisfying the triangular condition $\mathbf{k}_i = \mathbf{k}_{i+r} \pm \mathbf{k}_{i+s}$, provided that $|i + r| < N$ and $|i + s| \leqslant N$, with $(r, s) \in \mathbb{Z}$.

Similar systems have been extensively studied in the past, especially for fluid and magneto-hydrodynamic turbulence [10, 33, 36–39]. By following the same direction, here a five modes truncated model $\mathcal{T}_5(u,v,\eta)$ have been numerically investigated, by selecting the following wave–vectors $\mathbf{k}_i = k_0\,\mathbf{n}_i$: $\mathbf{n}_1 = (0, 1)$, $\mathbf{n}_2 = (1, 1)$, $\mathbf{n}_3 = (1, 2)$, $\mathbf{n}_4 = (2, -1)$, and $\mathbf{n}_5 = (3, 0)$, which satisfy the triangular relations $\mathbf{k}_1 \equiv \mathbf{k}_3 - \mathbf{k}_2$, $\mathbf{k}_2 \equiv \mathbf{k}_3 - \mathbf{k}_1 \equiv \mathbf{k}_5 - \mathbf{k}_4$, $\mathbf{k}_3 = \mathbf{k}_2 + \mathbf{k}_1$.

By exploiting the conditions $u_{-i}(t) = u_i^\star(t)$, $v_{-i}(t) = v_i^\star(t)$, $\eta_{-i}(t) = \eta_i^\star(i)$ on the complex conjugates, and by expanding the sums in nonlinear terms, the dimensionless RSW model is reduced to an autonomous dynamical system describing the dynamic of two interacting triads, containing, both, local and non-local interactions. The geometry of the system is sketched in Fig 1. The local triad is defined by a coupling between nearest wavevectors fulfilling the condition $D = C\sqrt{ab}$, with $a = b$, and $C = \sqrt{2}$ (Fig 1), while the non-local triad is composed by "more distant wavevectors" with the condition $a \neq b$ and $C \neq \sqrt{2}$, respectively. The first triad

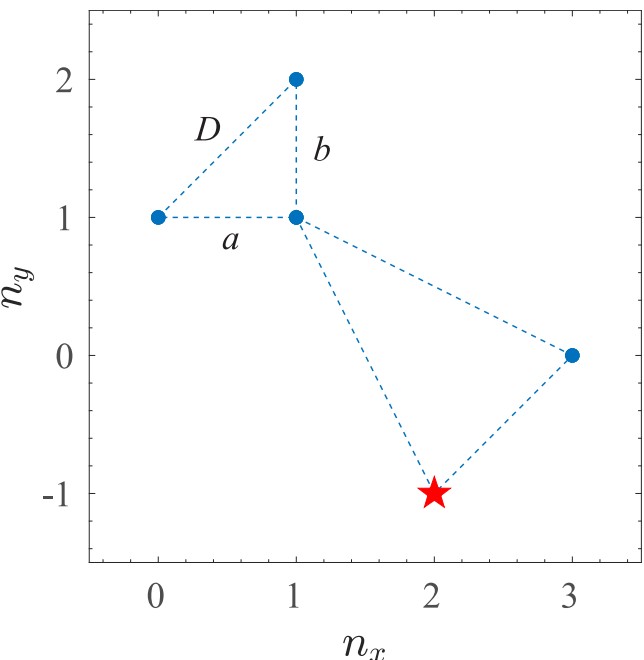

**Fig 1. Wave vectors configuration on the Fourier $k_x$, $k_y$ plane used in the $\mathcal{T}_5$ system, where the local triad consists of modes $k_1$, $k_2$, $k_3$ ($C = \sqrt{2}$), and the non–local one is composed by $k_2$, $k_4$, $k_5$ ($C \neq \sqrt{2}$).** The red star represents the forced mode $k_4$.

composed by wave vectors $\mathbf{k}_1$, $\mathbf{k}_2$, and $\mathbf{k}_3$ represents the prototypical local interactions, while the second triad composed by $\mathbf{k}_2$, $\mathbf{k}_4$, and $\mathbf{k}_5$ is relative to non–local interactions, since the "distant" modes $\mathbf{k}_4$, and $\mathbf{k}_5$ are able to communicate, in terms of energy transfer [40] or information exchange [41, 42] with $\mathbf{k}_1$, $\mathbf{k}_3$ via the mode $\mathbf{k}_2$, which represent the junction point between the two triads. A schematic of the wave vectors configuration, composing the two triads on the $k_x$, $k_y$ plane in the Fourier space, is reported in Fig 1, with the external forcing, represented as a red star localized on the mode $k_4$. With this choice, the energy is injected into the system via the non–local triad on a scale $k_f = k_0\sqrt{5}$, and subsequently transferred to smaller $k$ through an inverse cascade process [43].

Within this framework, the temporal evolution of the autonomous dynamical system $\mathcal{T}_5$ is described by a set of 15 complex ordinary differential equations, which, for the sake of clarity and readability of the text, are reported in the Section S1.1 in S1 File.

## 4 Numerical results

After the proper non-dimensionalization, the Rossby number is Ro = $10^3$, while the Reynolds number has been chosen of the order of Re = $5 \times 10^2$, while, for simplicity, the length scale of the system $L$ is considered ten times the depth (10$H$). This choice limits the possible wave periods observable in the system by selecting only periods smaller than $f^{-1}$, thus making the rotation a constant external energy source, acting equally on all scales. The value of $L$ is mainly related to the range of wavevectors considered in the domain, and which are actively involved in defining the "strength" of the nonlinear interactions of the truncated system (10)–(12). Selecting a small $L$ implies observing large wavevectors, while a large $L$ value selects small wavevectors. In light of this, by selecting a too-large $L$, the contribution of the nonlinear term

to the dynamic of the system would become too small, with the consequent inability to observe any transition to chaos in the system. For this reason, a value $L = 10H$ was considered here since it represents a fair compromise between the shallow-water approximation and the possibility of observing the nonlinear dynamics with only a few modes. It is worth noting that by varying the Reynolds number Re the global route among the various dynamics is not altered, but rather, the only difference is that the various regimes are shifted to a lower and narrower range of the external forcing values, which makes impossible a clear distinction between such regimes.

Each numerical run is integrated using a classical fourth-order Runge-Kutta time marching scheme [44], with a fixed nondimensional timestep $\Delta t = 0.125$. The quantity $T\,\mathrm{Ro}^{-1}$ (where $T = t/t_w$) has been used as a temporal unit in the various figures in order to represent the time variables in the unit of the rotation time of the system. All results presented in the following sections are relative to 150 rotation times enclosed in the interval $600 \leqslant T\,\mathrm{Ro}^{-1} \leqslant 750$.

Each run is initialized with the same random initial condition

$$\{u_k(0), v_k(0), \eta_k(0)\} \in \mathbb{C}^3,$$

where the real part of each coefficient is normally distributed with zero mean and standard deviation $\sigma_0 = 0.025$, and imaginary part uniformly distributed in the interval $[0, 2\pi]$. Despite the crude simplification, this $\mathcal{T}_5$ model possesses rich and various dynamical behaviours, which can be explored by varying the control parameter $\mathcal{C} := |F_0|\mathrm{Re}$ in the range $\mathcal{C} \in [0.15, 8.5]$. The control parameter was defined in this form in order to map the various dynamics arising in the $\mathcal{T}_5$ model in terms of fractions of the Reynolds number Re.

As an initial step, the study concentrates on ensuring the conservation of the two invariants in scenarios where both $\mathcal{D}$ and $\mathcal{F}$ are equal to zero. To check their conservation is sufficient to limit the infinite summation in relation (5) and (6) to the finite number of modes evolved via the $\mathcal{T}_5$, then the two quantities, say the total energy density $\mathcal{E}(t)$, and the potential vorticity $\mathcal{Q}(t)$ have been calculated in the physical space, in order to avoid the triple product $hu^2$, $hv^2$ in the Fourier space. Fig 2 reports the temporal behaviour of the two invariants for each triad

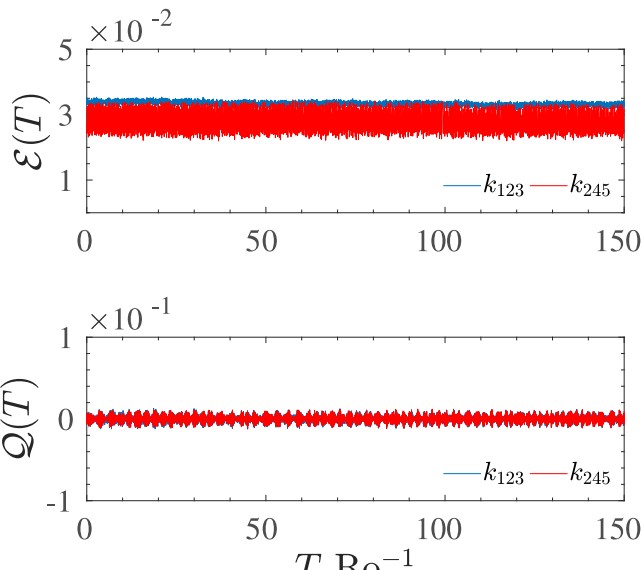

**Fig 2.** Upper panel: Temporal evolution of the total energy density $\mathcal{E}(T)$ for the local triad $k_{123}$ and for the non–local triad $k_{245}$, respectively. Lower panel: temporal evolution of the potential vorticity $\mathcal{Q}(T)$ evaluated for the two triads.

composing the $\mathcal{T}_5$ system, obtained for an unforced inviscid case. A good conservation is achieved for both invariants, with fluctuations characterized by an amplitude of the order of $\mathcal{O}(2)$. However, the amplitude of these fluctuations is reduced if a higher-order time marching scheme is used or if the integration timestep $\Delta t$ is reduced. The small difference observed among the amplitudes of the two triads is due to the different modulus $k_i$ of the wave vectors composing them. In particular, the oscillations in the non-local triad are favoured over those observed in the local triad since smaller scales (higher wave vectors) are more easily destabilized than larger ones (smaller wave vectors). As in direct numerical simulations, these small fluctuations are greatly reduced by averaging the effects of multiple resonant triads. In fact, in a direct simulation, the triangular condition $\mathbf{k} = \mathbf{p} + \mathbf{q}$ depends on the sum of infinite other wave vectors, so the fluctuations of every single triad will be an averaged (cumulative) effect of all the other triads present in the system, for which the condition is satisfied.

To better understand the temporal dynamics of the various scales involved in the $\mathcal{T}_5$ system, as an example, in Fig 3 is reported the temporal behaviour of real part of the various coefficients the $u$-component, say $\mathcal{R}\{u_k(t)\}$ (with $k \in [1, 2, \ldots, 5]$), in a forced-dissipated run, and for a time interval proportional to ten rotations time. The two panels in Fig 3 show two different states of the system, depicting the transition from a smooth periodic motion observed at $\mathcal{C} = 0.17$ (Fig 3 upper panel) to a chaotic state of the motion recorded for $\mathcal{C} = 0.21$ (Fig 3 lower panel), respectively. In both cases, all modes oscillate with an amplitude greater than zero, whose values are strictly related to the modulus of the associated wave vector $k$. Obviously, the mode that exhibits the minimum amplitude is the first wave vector $k_1$, whose amplitude is of the order of $|\mathcal{R}\{u_1\}| \approx 10^{-6}$ and is represented in Fig 3 by the purple line oscillating near the zero value, since this wave vector is the most difficult to destabilize in contrast to mode $k_5$ which oscillate with the maximum amplitude $|\mathcal{R}\{u_5\}| \approx 10^{-2}$.

In order to simplify the description of system dynamics, rather than analyze the dynamics of every single mode $k$ composing the $\mathcal{T}_5$ system, here the attention has been focused on four "global state variables", which are defined as $E_\xi := \sum_j |\xi_j|^2$ (being $\xi = u, v$), $E_k := (1/2)(E_u + E_v) \equiv$

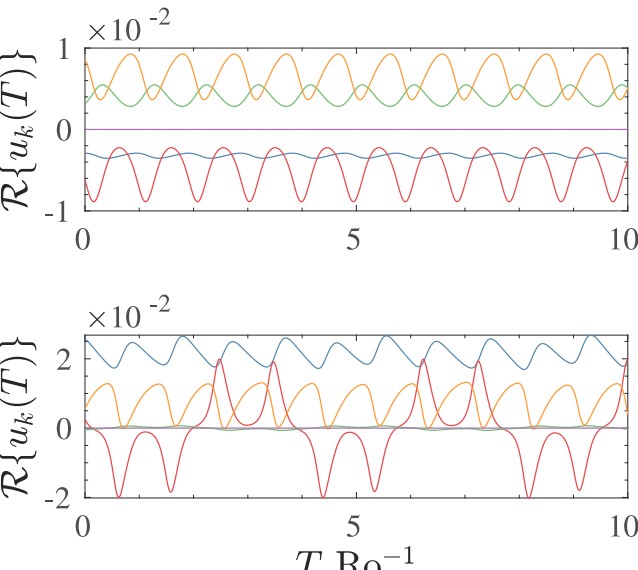

**Fig 3. Temporal evolution of the real part of $u_k(t)$ ($k \in [1, 2, \cdots, 5]$) component of the $\mathcal{T}_5$ system for two distinct values of the control parameter: $\mathcal{C} = 0.17$ (periodic state), and $\mathcal{C} = 0.21$ (chaotic state).** The amplitude of the oscillations is strictly dependent on the wave vector $k_i$. The minimal amplitude is recorded for $k_1$, $\mathcal{R}\{u_1\} \approx 10^{-6}$.

(1/2) $\sum_j(|u_j|^2 + |v_j|^2)$, and $E_\eta := \sum_j|\eta_j|^2$, representing a pseudo-kinetic and pseudo-potential energy, respectively. An example of the temporal dynamic of such four quantities, for various values of the control parameter $\mathcal{C}$, is reported in Fig 7 and will be discussed in detail in the following subsections.

It is interesting to note that when the forcing is applied on both velocity components $\{u(\mathbf{r}, t), v(\mathbf{r}, t)\}$, independently from the amplitude of the control parameter $\mathcal{C}$, the system always reaches a state characterized by a periodic motion, as the one depicted in the upper panel of Fig 3, while the characteristic oscillations period decreases as the control parameter $\mathcal{C}$ increases.

## 4.1 A global view of the dynamics of the system

A macroscopic view of the behaviour of the $\mathcal{T}_5$ system is depicted in Fig 4, where the bifurcation diagrams obtained by sampling all extreme values of the four state variables, as a function of the control parameter $\mathcal{C}$. Interestingly, all state variables share the route to chaos and the same bifurcation process. The only difference that is noticeable is their magnitude. For very weak values of the forcing (not shown here), the system has a trivial fixed point where all coefficients $(u_k, v_k, \eta_k) = \vec{0}$. This state is destabilized, as $\mathcal{C}$ increases, through a pitchfork bifurcation, towards a new steady state $E_u = E_v = E_k = E_\eta \neq 0, \forall t \geqslant 0$. For a fixed threshold value of the control parameter: $\mathcal{C} \approx 0.1605$, the system is, in turn, destabilized through a Hopf bifurcation, maintaining oscillating periodic solutions in a range of control parameter values $0.1605 \lesssim \mathcal{C} \lesssim 0.199$. This first transition represents a universal behaviour which is always found in low-dimensional fluids mechanics models [10, 33, 36, 45].

In the range $0.2 \lesssim \mathcal{C} \lesssim 0.235$, the system enters a complex region characterized by alternation between chaotic states and laminar periodicity windows. Beyond the chaotic region, the

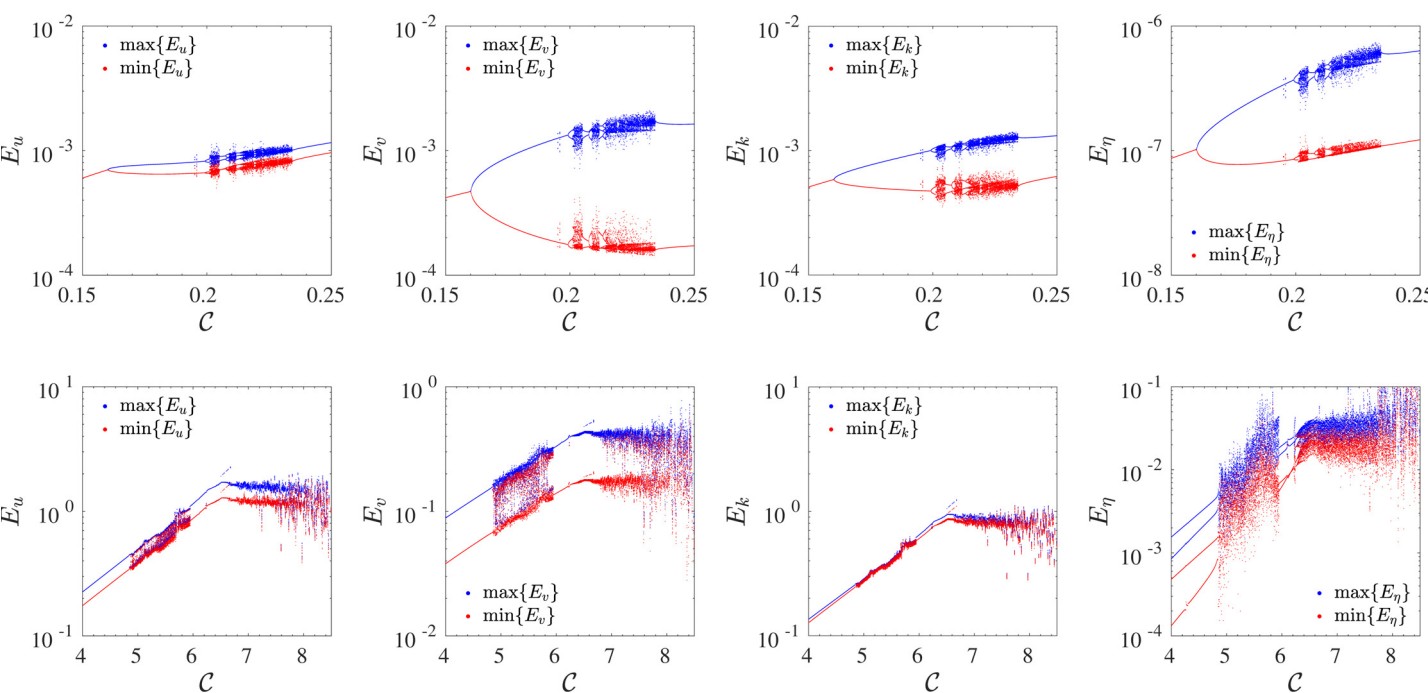

**Fig 4.** First row: bifurcation diagrams for all four state variables $\{E_u, E_v, E_k, E_\eta\}$ in a range of control parameter $0.15 \leqslant \mathcal{C} \leqslant 0.25$, depicting the low forcing regime of the system. Second row: bifurcation diagrams for the second range of control parameter $4 \leqslant \mathcal{C} \leqslant 8.5$, relative to the moderate/high forcing regime of the system. In each diagram, the upper branch (blue dots) represents the maximal values recorded for each state variable, and the lower branch (red dots) is relative to their minimum values.

system falls in a large periodicity window spanning the range $0.235 < \mathcal{C} \lesssim 4.8$, characterized by a continuous amplitude growth of all state variables, then a secondary transition to a continuous chaotic state is observed ($4.8 < \mathcal{C} \lesssim 5.9$) followed again by small periodicity windows up to $\mathcal{C} \lesssim 6.2$. Above this threshold, another chaotic region is observed. However, when the control parameter approaches the value $\mathcal{C} \approx 6.5$, the amplitude growth comes to a halt, settling at a stationary value, different for each state variable. For $\mathcal{C} > 8.6$, the system turns unstable since, under the effects of the strong external forcing, the maximal amplitude of the displacement may become much greater than the reference height $H$: $\max\{\eta(\mathbf{r}, t)\} \gg H$.

As the control parameter $\mathcal{C}$ increases, the average oscillation period $\tilde{\mathcal{T}}$, measured as the time difference between two local extreme points (maxima or minima) recorded in the temporal dynamics of the state variables $\{E_u, E_v, E_k, E_\eta\}$, rapidly decrease, passing from one rotation time for lower $\mathcal{C}$ to fractions of the rotation time for higher values of the control parameter. An example relative to the state variable $E_u$ is reported in Fig 5. The system reaches a value approximately proportional to a unitary rotation time, $\tilde{\mathcal{T}} \, \mathrm{Ro}^{-1} \simeq 1$ when the system approaches the first chaotic region around $\mathcal{C} \approx 0.2$, then smoothly decreases, indicating the importance of the rotative effects in the transition to the chaotic regime of the system, and thus how the rotative effects could potentially affect the turbulent behaviour in the RSW. The smooth shrinking of the average period continues until the threshold value $\mathcal{C} \approx 4.9$, where $\tilde{\mathcal{T}}$ exhibits an abrupt change that is maintained up to $\mathcal{C} \gtrsim 5$. Such range represents a secondary chaotic zone before the system reaches the plateau for $\mathcal{C} \approx 6.5$. In this region, the system presents very fast oscillations, with a characteristic period proportional to $\tilde{\mathcal{T}} \, \mathrm{Ro}^{-1} \approx 5 \times 10^{-3}$ rotation times. Another abrupt change is also observed in the third chaotic zone for $\mathcal{C} \gtrsim 6.5$, where the average period reaches similar values but with a smoother path depending on $\mathcal{C}$. The same behaviour is also observed for other state variables of the system.

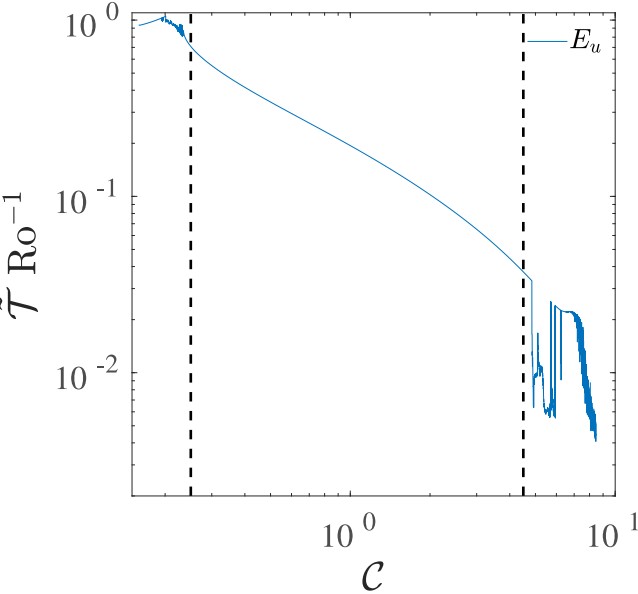

**Fig 5. Evolution of average oscillation period $\tau_u$ relative to the state variable Eu as a function of the control parameter $\mathcal{C}$.** The parameter $\tilde{\mathcal{T}}$ is defined as the time difference between two local maxima or minima in the temporal dynamic of the state variable. The dashed lines have been used to separate three principal dynamical behaviours of the $\tilde{\mathcal{T}}_5$ system: low–forcing zone, periodic zone, and high forcing zone.

Since most studies of the RSW system are carried out in the physical space, in order to better understand the various dynamical regimes observed in the $\mathcal{T}_5$ system, and possibly establish a link with other classical works, in Fig 6 are reported the Hovmöller diagram of displacement $\eta(x, t)$. The field has been reconstructed in the physical space on a square periodic domain of size $L$ with a resolution of $N_p = 256$ grid points; this choice ensures that large-scale structures (small wave vectors) can be reconstructed without aliasing problems and observed correctly in the physical space. Each diagram represents a cut of the physical space reconstruction of $\eta(\mathbf{r}, t)$ along the principal diagonal of the domain.

In particular, in the four panels of Fig 6 are reported two cases where the system exhibits a periodic oscillating behaviour with $\mathcal{C} = 0.19$ and $\mathcal{C} = 4$ (Fig 6 first and second panel), and two snapshots of the system in the chaotic state, observed for $\mathcal{C} = 0.23$ and $\mathcal{C} = 7$ (Fig 6 the third and the fourth panels). The shrinking of the average period is also evident, especially by observing the first two panels of Fig 6, for the periodic case. In fact, for $\mathcal{C} = 0.19$ (Fig 6, first panel), the system exhibits a sort of "breathing" pattern, characterized by the periodic alternation of large-scale structures forming on a time scale proportional to $T \, \mathrm{Ro}^{-1} \approx 1$, which evolve into much faster dynamics, characterized by $T \, \mathrm{Ro}^{-1} \approx 1$ when the system is driven with a control parameter $\mathcal{C} = 4$ (Fig 6, the second panel). The chaotic case $\mathcal{C} = 0.23$ (Fig 6, the third panel), is characterized by large-scale structures, with superimposed formation of smaller secondary structures. The "breathing" structure is still present but is composed of the alternation of two recurrent patterns: one composed of the same large-scale structures observed in the

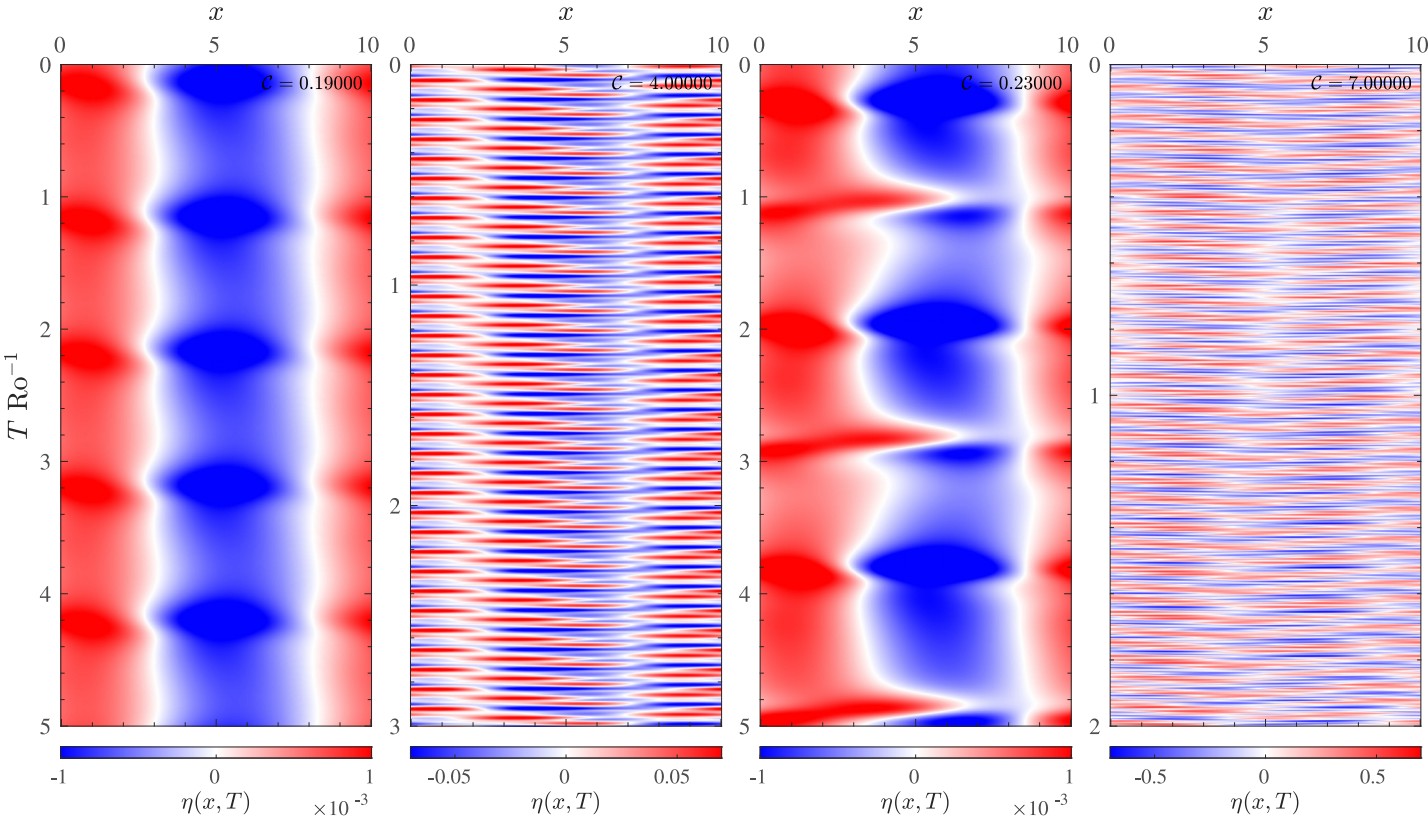

**Fig 6. Hövmoller diagram for the surface displacement $\eta(r, t)$ reconstructed in the physical space, for different values of the control parameter $\mathcal{C}$, depicting the system during a periodic solution (first and second panel), and during the chaotic regime (third and fourth panel).** The first and the third panels are relative to the low-forcing regime, while the second and fourth represent the high-forcing regime. The diagram was obtained by a field cut along the principal diagonal.

periodic case and a composition of smaller-scale structures which periodically break the large-scale structure. Each pattern is reproduced with a period $T\,\mathrm{Ro}^{-1} \approx 2$. Finally, for higher values of $\mathcal{C}$ (Fig 6, the fourth panel), the system is characterized only by a complex pattern evolving on a very fast timescale, moreover the recurrence in the pattern is very hard to find. However, it is interesting to note that the frequency at which the structures evolve is also related to the amplitude of the field $\eta(\mathbf{r}, t)$, in fact, as $\mathcal{C}$ increases, the $\eta$ field-related forcing, represented by the linear term in the RSW system, becomes increasingly important. In light of this finding, the complex dynamic of the $\mathcal{T}_5$ model can be separated into three different regimes, which can be analyzed in detail separately:

1. A low-forcing stage characterized by periodic/quasi-periodic and chaotic oscillations,

2. A quiescent zone characterized only by periodic oscillations with shortening of the characteristic period,

3. A moderate/high-forcing stage where a sudden transition to chaotic oscillations happens.

The three zones have been highlighted in Fig 5, which appears separated by two vertical lines. In the sequel, the attention will be focused principally on the first and third regimes.

## 4.2 Route to chaos in the low–forcing regime

As the parameter $\mathcal{C}$ increases, the system enters different regimes characterized by increasing complexity after the main bifurcation. The system oscillates with structures characterized by an increasing number of harmonics or develops a large number of multi-periodic structures up to a limit at which they vanish and merge in a disordered intermittent regime for $\mathcal{C} \gtrsim 0.21$. Such behaviour is observed in all state variables. The evolution of such multi-periodic dynamics is reported in the first two rows of Fig 7, for $E_u$ (the first row) and for $E_\eta$ (the second row),

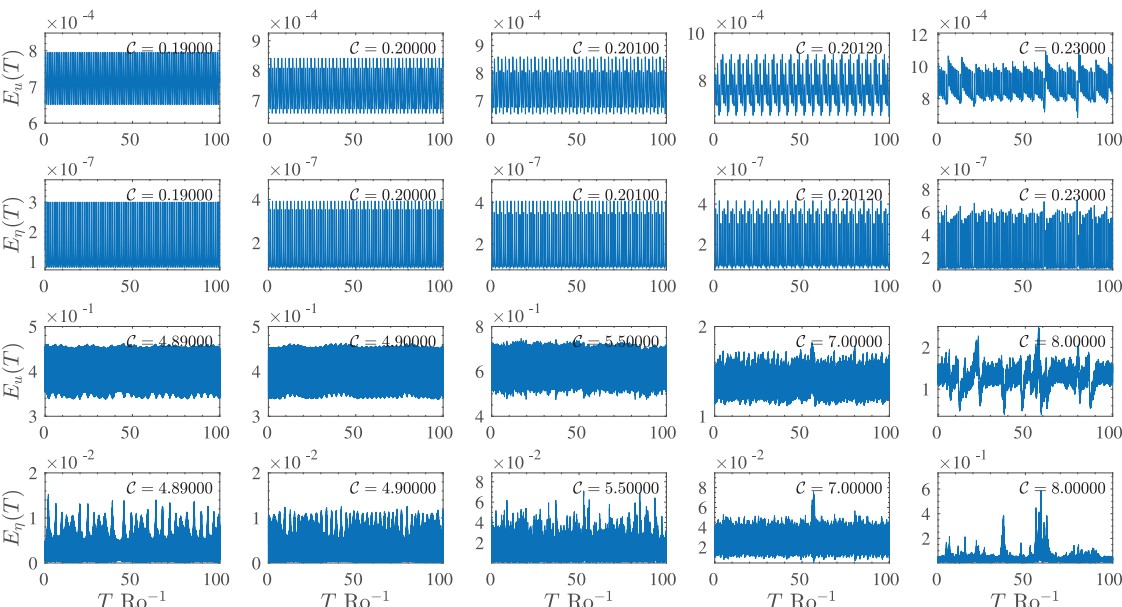

**Fig 7. Temporal evolution of the state variables $E_u$, $E_\eta$ as the control parameter $\mathcal{C}$ increases.** The first two rows show the evolution of the dynamics and the transition from the periodic regime to the multi-periodic regime that finally results in the chaotic regime, for $0.19 \leqslant \mathcal{C} \leqslant 0.23$ (low-forcing regime). The third and fourth rows show the evolution of high–forcing dynamics, where the system shows a fast transition from a periodic regime to an intermittent regime to finally reach a regime characterized by strong spikes.

respectively, in a time interval proportional to 100 rotation time. This type of transition has been observed in several systems and is generally accompanied by a doubling of the oscillation period, and such doubling is translated in phase space trajectories as a splitting of the system's orbits.

A visual example of the orbit splitting is depicted in the first two panels of Fig 8 (first row), where the phase space of the system is composed by the state variables $\{E_u, E_v, E_\eta\}$. The initial periodic orbit splits into two characteristic orbits when the control parameter reaches the value $\mathcal{C} \approx 0.20$. As $\mathcal{C}$ increases, the splitting continues developing more and more orbits (Fig 8, the first row, the first and the second panel) in a continuous sequence until the system fully transits to a chaotic dynamic developing multiple strange attractors (Fig 8, first-row third panel). During the temporal evolution, the dynamics evolves on all axes representing the state variables (Fig 8, first row). As the control parameter $\mathcal{C}$ increases, the dynamics occurs on multiple planes intersecting the phase space at different angles, and this angle of intersection varies as time progresses. This effect is particularly evident in chaotic regions (Fig 8, the first row, the third panel) where the dynamics evolve in both vertical ($E_u, E_\eta, E_v, E_\eta$) and horizontal planes ($E_u, E_v$) or in more complex situations involving all possible angles.

The splitting effect is clearly observable by constructing the classical Poincaré return maps for the three variables composing the phase-space (Fig 9). Here, the maps have been constructed by recording all extreme points observed during the temporal dynamic of three state variables: ($E_u, E_v, E_\eta$), respectively, and relating the $i$-th maximum with the following ($i + 1$)-th one. Starting from the periodic state ($\mathcal{C} = 0.19$), as the control parameter increases, the return

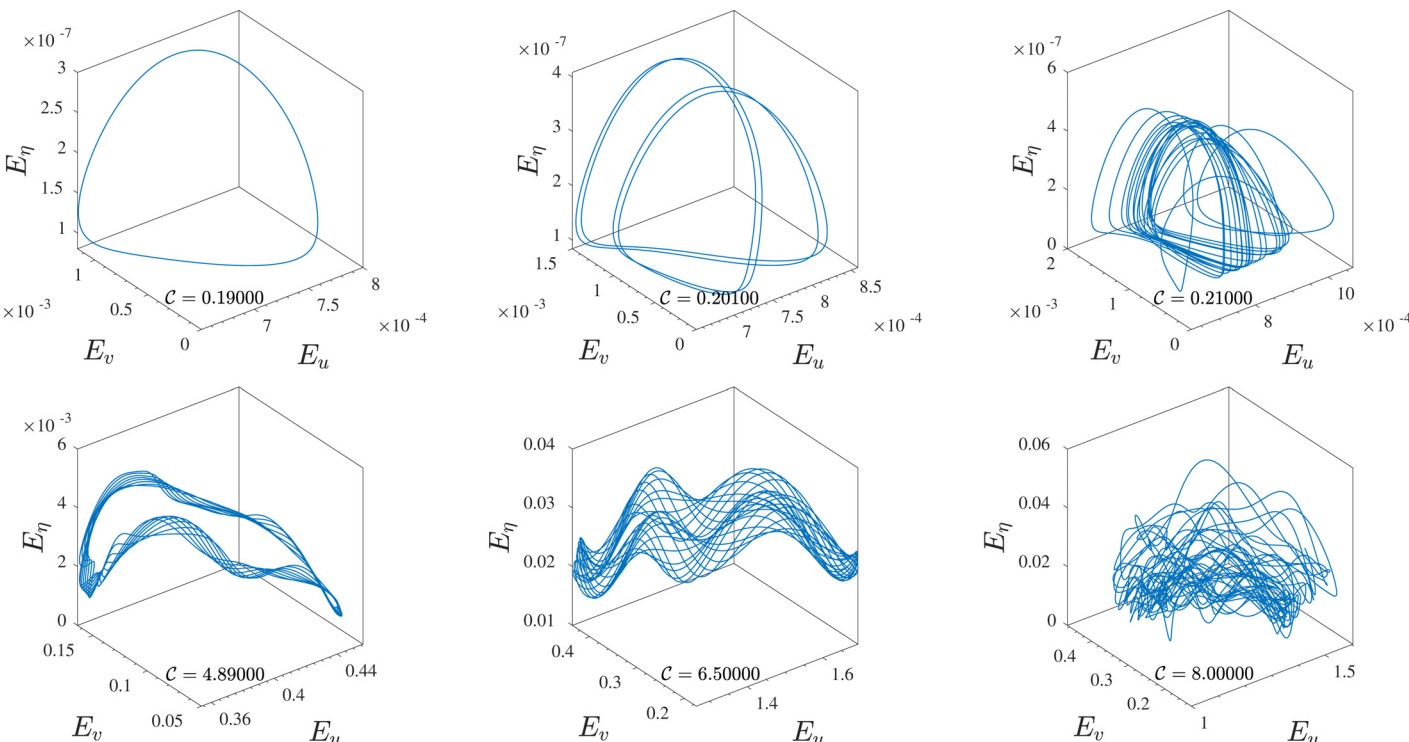

**Fig 8. Evolution of the dynamic in the phase-space composed by the state variables $\{E_v(T), E_v(T), E_\eta(T)\}$.** First row: dynamics recorded in the low-forcing regime depicting the transition across the various dynamical regimes of the system: periodic (left panel), multi-periodic (central panel), and chaotic (right panel), where multiple strange attractors appear. Second row: dynamics recorded in the low-forcing regime, contrary to the previous case, the dynamics occurs predominantly on two state variables ($E_u, E_v$), while the third, $E_\eta$, weakly oscillates around an average value (left and central panel). The complete exploration of the phase space is observed only for very high values of $\mathcal{C}$ (right panel).

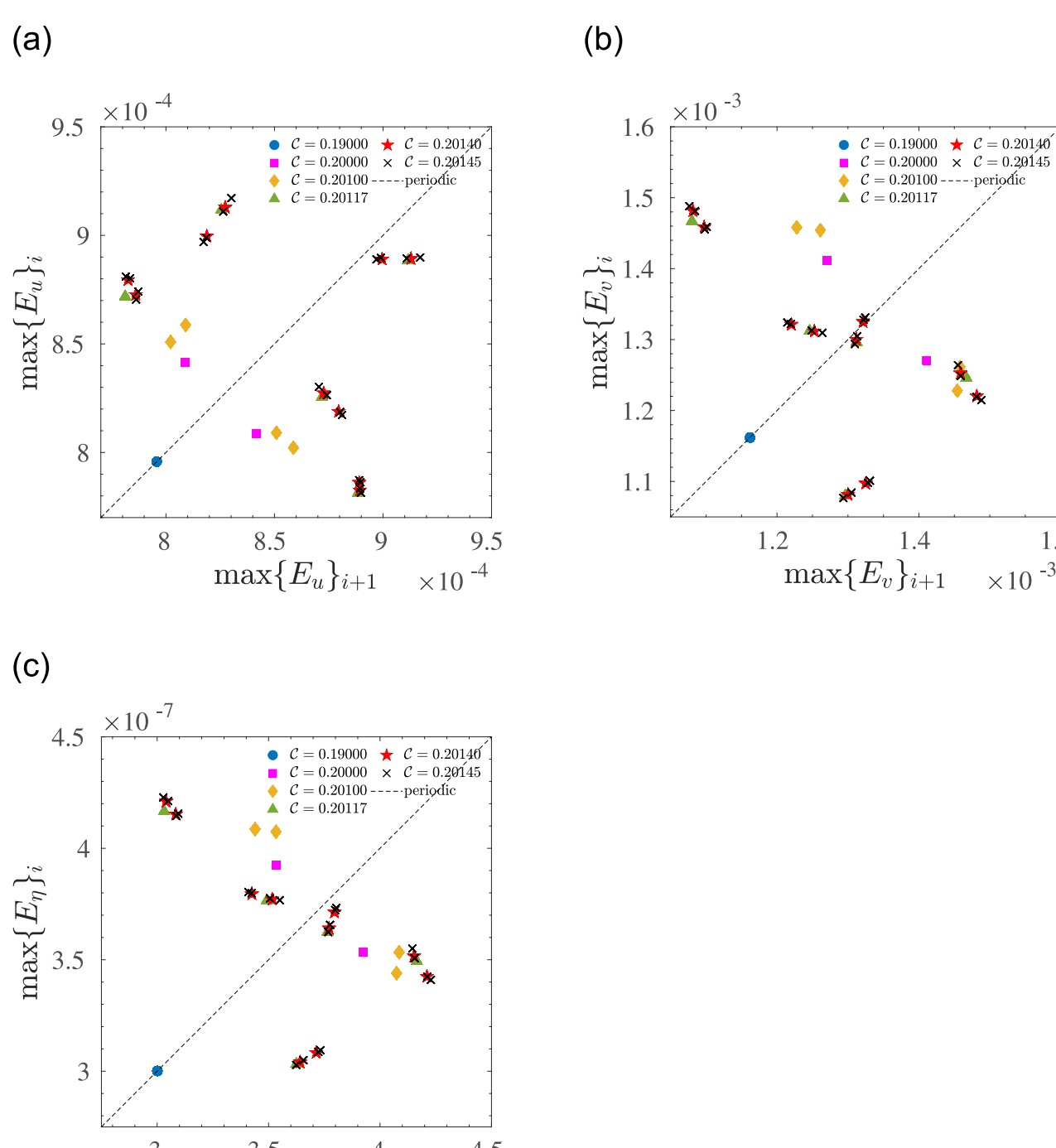

**Fig 9. Poincaré return maps as a function of the control parameter $\mathcal{C}$, showing the transition from periodic (circles) to a multi-periodic solution (other symbols, see legend for colour coding) for three state variables: ($E_u$, $E_v$, $E_\eta$), respectively.** The doubling period sequence breaks at $\mathcal{C} = 0.20117$, where five points appear on the plane, then the sequence restarts and each point is split in two. The dashed line represents the periodic behaviour of state variables.

**Table 1. Control parameter $\mathcal{C}$ values associated to the first six bifurcation points observed in the evolution of $\mathcal{T}_5$ system with complex variables, Eqs (10)–(12), and the local Feigenbaum constant.**

| $i$-th Bifurcation point | $\mathcal{C}_i^\star$ | $\delta$ |
|---|---|---|
| 1 | 0.19350 | |
| 2 | 0.20000 | 6.5 |
| 3 | 0.20100 | 5.9 |
| 4 | 0.20117 | 0.7 |
| 5 | 0.20140 | 4.6 |
| 6 | 0.20145 | |

maps evolve accordingly for each subsequent bifurcation, with an increasing number of points appearing on the plane, composed, here, by the extreme points of the state variables [10]. By analyzing in more detail those maps, it is possible to note that each intersection is followed by two points, or in other words, the state described by the intersection with $\mathcal{C} = 0.20$ is composed only of two points on the plane (Fig 9, magenta squares), and each point is supplemented by two more intersections observed at $\mathcal{C} = 0.201$ (Fig 9, yellow diamonds). Interestingly, such cascade of period doubling vanishes for $\mathcal{C} = 0.20117$, where there is no longer a doubling of orbits, but rather a net split is observed that opens five distinct orbits (Fig 9, green triangles). However, after this point, the sequence of doubling is restored. In fact, for each intersection observed at $\mathcal{C} = 0.20117$, two more periods appear for $\mathcal{C} = 0.20140$, and each one is split in two more at $\mathcal{C} = 0.20140$, (Fig 9, red stars).

The sequence of various bifurcations is compatible with the classical Feigenbaum picture [46], where the chaotic regime is reached as a sequence of infinite bifurcation points, or, in other words, in a continuous splitting of the orbit in the phase-space. In such case, a fixed ratio for the various bifurcation points holds:

$$\delta_\infty = \lim_{i \to \infty} \frac{\mathcal{C}_i^\star - \mathcal{C}_{i-1}^\star}{\mathcal{C}_{i+1}^\star - \mathcal{C}_i^\star} \simeq 4.6692\dots \, , \tag{13}$$

being $\mathcal{C}_i^\star$ the value of the control parameter at the $i$-th bifurcation point. By taking into account the bifurcation points reported in Table 1, the value of $\delta$ we obtain at the bifurcation point $\mathcal{C}_5^\star$ converges to a value compatible with the Feigenbaum $\delta_\infty$ constant. After the sixth bifurcation, the system shows a transition to a chaotic state, and further bifurcations have not been easily recognized. This route has also been observed in other systems, such as the case of two-dimensional Navier-Stokes equations [47], or the classical Lorenz system [48]. The occurrence of the Feigenbaum transition in the RSW system for small $\mathcal{C}$, analogous to five-modes truncation of the 2D Navier-Stokes equations, could be due to the fact that in this regime nonlinear interactions are mainly dominated by the inertial term because the potential energy is smaller in comparison with the kinetic energy $g\eta^2 \ll (u^2 + v^2)\eta$ (Figs 4 and 7).

## 4.3 Secondary transition in the high–forcing regime

Contrary to what occurs in the low $\mathcal{C}$ case, the second transition to the chaotic regime is instantaneous and impulsive. When the control parameter exceeds the second threshold (see section 4.1), the system immediately shifts on a chaotic orbit, characterized by a higher magnitude of the state variable $E_\eta$, which, in this case, begins to play a more important role in the overall process. In other words, the coupling effects of turbulence with waves within the system become evident, as demonstrated by the strong amplitude increase of the pseudo-potential energy (Fig 7, last row).

A substantial difference from the previous case can be easily noted by looking at the last two rows of Fig 7. In fact, as the control parameter increases, the dynamics of the systems present an increasingly disordered behaviour, characterized by the appearance of intermittent structures, whose amplitude is dependent on the value of $\mathcal{C}$. These structures initially have an almost periodic pattern over time (Fig 7, rows three and four, first two panels) but are characterized by variations in amplitude, which is especially evident in the state variable $E_\eta$ (Fig 7, the fourth row). As $\mathcal{C}$ increases, the quasi-periodicity tends to disappear, giving way to the formation of a strongly impulsive spike pattern with extreme amplitudes, as reported in the last two panels of Fig 7, the rows three and four.

The formation of these structures may indicate that the free surface $\eta(\mathbf{r}, t)$ is actively contributing to the turbulent dynamics of the system, acting as a forcing on the velocity, or in other words, the forcing $\mathcal{F}_u$ on velocity creates waves that during their propagation act in turn as a forcing on $\mathbf{u}(\mathbf{r}, t)$.

By analyzing the trajectories in the phase-space composed of the state variables $(E_u, E_v, E_\eta)$ (Fig 8 second row), a second substantial difference can be inferred with respect to the lower forcing regime. In this case, the phase space is not explored entirely in three dimensions, but rather, the trajectories remain constrained to evolve in planes almost parallel to the plane composed of the variables $(E_u, E_v)$. In other words, the dynamics evolve in planes that intersect the phase-space at around a mean value of $E_\eta$, and this effect becomes increasingly evident as the control parameter $\mathcal{C}$ increases.

## 4.4 Rossby number dependence

As known, the dynamic of the RSW is strongly affected by Coriolis forces for small Ro. Specifically, waves irreversibly impact the hydrostatically balanced flow, especially in the small Rossby number regime, where the flow can be easily perturbed due to the presence of high energy waves [49], while a large Ro implies that inertial and centrifugal forces dominate. In light of this, a further length scale can be defined as $L_R := \sqrt{gH}f^{-1}$, the so-called Rossby deformation radius, which represents the horizontal length scale over which the geostrophic adjustment holds [2] (balanced flow regime). When the domain length $L$ is small with respect to $L_R$, say $L \ll 2\pi L_R$, the rotation effects may be weaker.

In light of this, analysing how the rotation affects the chaotic dynamic is interesting. For this reason, two control parameter values have been selected: $\mathcal{C} = 0.23$ and $\mathcal{C} = 7$. These two values were chosen specifically as they represent the RSW system in two chaotic cases (*i.e.* the first related to the low-forcing regime, and the second relative to high–forcing regime), while the Rossby number Ro was varied in the range Ro $\in [0.1, ..., 500]$.

For the low-forcing regime (Fig 10, the first row, $\mathcal{C} = 0.23$), the dynamics is modified when Ro $\leqslant 1$, where chaotic oscillations are suppressed in favour of a periodic dynamics with mean period $\tilde{\mathcal{T}}$ Ro$^{-1} \approx 1.5 \times 10^3$ for Ro = 1, while the system falls on the steady state (or very long period oscillations) for Ro = 0.1. On the other hand, when Ro > 1, the behaviour is chaotic, and again characterized by a narrowing of the mean period, passing from $\tilde{\mathcal{T}}$ Ro$^{-1} \approx 90$ for Ro = 10 to $\tilde{\mathcal{T}}$ Ro$^{-1} \approx 1.7$ for Ro = 500, and $\tilde{\mathcal{T}}$ Ro$^{-1} \approx 0.9$ for Ro = 1000.

In the high–forcing regime (Fig 10, the second row where $\mathcal{C} = 7$), waves severely modify the flow dynamics. However, the system falls again on the steady state for very low Rossby number, *e.g.* Ro = 0.1, but an intermittent chaotic behaviour is observed for Ro = 1 (Fig 10, the first panel of the second row). Again, this intermittent regime disappears for moderate Rossby numbers (*e.g.* Ro = 10, Fig 10, the third panel of the second row), giving places to a periodic motion with characteristic period $\tilde{\mathcal{T}}$ Ro$^{-1} \approx 0.5$. Finally the chaotic behaviour is restored for higher Ro (Fig 10, last two panels of second row).

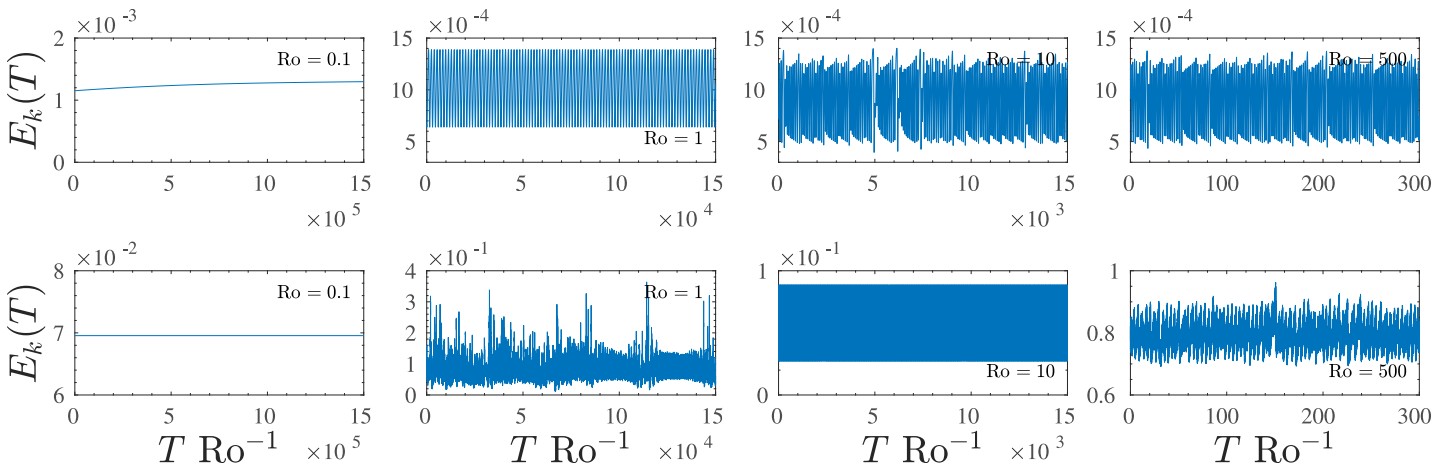

**Fig 10. Temporal evolution of the state variable $E_k(T)$, as a function of the Rossby number Ro for $\mathcal{C} = 0.23$ (first row) and $\mathcal{C} = 7$ (second row).** The RSW's dynamic is affected by the parameter Ro since its variation can suppress chaos or create a stronger intermittent chaotic regime.

## 5 Phase locking and phase dynamic in the truncated RSW system

Since Fourier coefficients of the RSW Eqs (10)–(12) are complex variables, the fields can be rewritten in terms of phases and amplitudes by defining $u_k := A_k e^{i\alpha_k}$, $v_k := B_k e^{i\beta_k}$ and $\eta_k := \Gamma_k e^{i\gamma_k}$. By using this representation, a new set of equations can be obtained describing the temporal evolution of both amplitudes and phases of the fields, reported for completeness in Section S1.2 in S1 File. The system evolves in the $(6 \times N)$-D phase space whose axes are formed by amplitudes and phases. When the fields are written in terms of phases and amplitudes, a new state variable appears; namely, the equations contain the three-phase coupling terms $\Phi_{\pm kpq}^{\alpha\beta\gamma}$, which can be written as:

$$\Phi_{\pm kpq}^{\alpha\beta\gamma} := \pm \alpha_k + \beta_p + \gamma_q. \tag{14}$$

This quantity satisfies the obvious parity relation $\Phi_{-kpq}^{\alpha\beta\gamma} = -\Phi_{k-p-q}^{\alpha\beta\gamma}$. Since the equations contain these quantities rather than the single phases, it is then interesting to investigate the dynamics of the three-phase coupling terms.

### 5.1 Invariant subspace

By analyzing the structure of the equations reported in Section S1.2 in S1 File, it is worth noting that the dynamics of the truncated low-dimensional system $\mathcal{T}_N(u, v, \eta)$, to any order $N$, can be split into two subsystems: $\mathcal{T}_N(u, v, \eta) = \mathcal{R}_N(A, B, \Gamma) \cup \mathcal{I}_N(u, v, \eta)$. This property is particularly interesting because the whole system can be reduced to a purely real system, say $\mathcal{I}_N(u, v, \eta) = \emptyset$, when the following conditions are satisfied:

1. The external forcing term is purely real: $\mathcal{I}\{\mathcal{F}(t)\} = 0$, $\forall t \geqslant 0$;

2. The relations: $\alpha_k = \pm\beta_k$ and $\gamma_k = \alpha_k \pm \pi/2$ holds among single phase variables;

3. The phases are locked in a way that the three-phase coupling terms satisfy the relations $\cos \Phi_{\pm kpq}^{\alpha\beta\gamma} = 0$, for all triad-phases involved in the equations.

These conditions define an "invariant subspace" for the dynamics of the system. In other words, when such relations are satisfied, the dynamics of the system occur only on the real

subspace $\mathcal{R}_N(A, B, \Gamma)$, implying that the dimensionality of the system is then reduced to ($3 \times$ $N$)-D. If the initial conditions of the system are constructed such that they satisfy the above conditions, then the system evolves for all times within the $\mathcal{R}_N(A, B, \Gamma)$ subspace. Numerical simulations (not shown here) show that the subspace $\mathcal{R}_N(A, B, \Gamma)$ is unstable, namely a small perturbation on $\mathcal{I}_N(u, v, \eta)$ is enough to destabilize the dynamics on the real subspace.

### 5.2 Phase dynamics and Phase-Transition Curve

In Fig 11 is reported the temporal evolution of $\alpha_k$ (upper row) and $\gamma_k$ (lower row), the dynamic of $\beta_k$ (central row) is similar to $\alpha_k$, the main difference found is that the oscillations do not occur on the same $k$ modes but on others, indeed in the left column Fig 11 (relative to the periodic regime of the RSW), in the the first and second panel the step-like structure passes from mode $k = 5$ (for phase variable $\alpha_k$) linked to the fastest mode, which oscillates periodically, to mode $k = 1$ (for phase variable $\beta_k$, linked to the slowest mode). Such inversion from the fastest to the slowest modes is observed in all dynamical regimes of the $\mathcal{T}_5$ system. The other phases

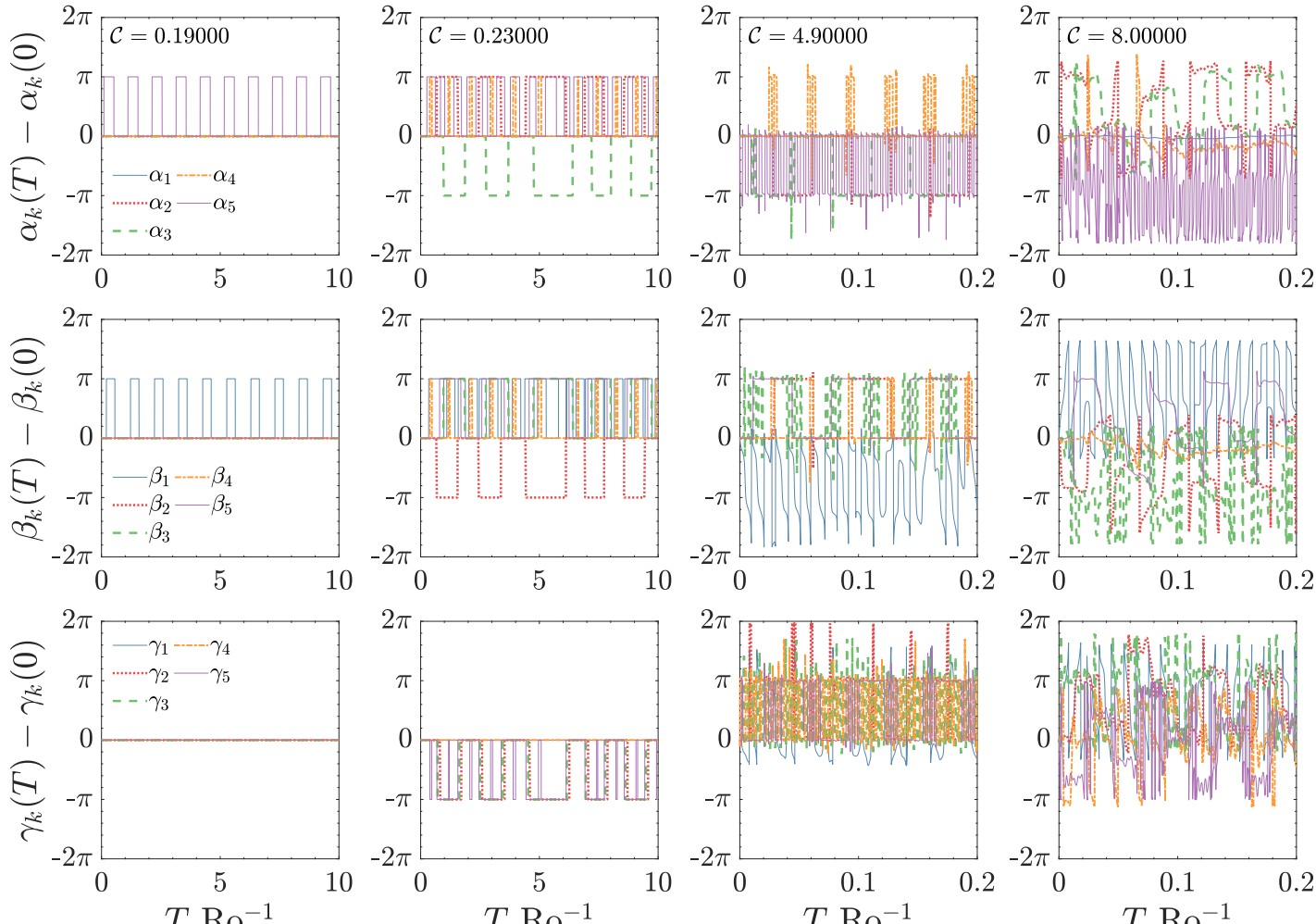

**Fig 11. Temporal evolution of the phase variables $\alpha_k(t)$ (the first row), $\beta_k(t)$ (the second row), and $\gamma_k(t)$ (the third row).** Each column represents a precise control parameter value $\mathcal{C}$. Moreover, the first two columns are relative to the low–forcing regime, specifically the periodic case ($\mathcal{C} = 0.19$) and the chaotic case ($\mathcal{C} = 0.23$); the third and fourth columns are relative to the different chaotic behaviour observed in the high–forcing regime, the increasing stage ($\mathcal{C} = 4.9$), and the plateau stage ($\mathcal{C} = 8.0$).

remain constant on the initial values. Interestingly, all phases $\gamma_k$ remain locked to the initial value, indicating that waves propagate at a constant speed when the system falls in this periodic state. The oscillating dynamics of phases, when the system reaches the chaotic state is rather peculiar, being formed by trains of periodic pulses of amplitude $\pm\pi$ (step-like locking), width $\delta\tau$ (representing the locking duration), and period $T_0$, which can be analytically defined by the following functional form:

$$\phi^{(k)}(t) = \pm\pi\frac{\delta\tau^{(k)}}{T_0^k} \pm 2\sum_n \sin\left(n\pi\frac{\delta\tau^{(k)}}{T_0^{(k)}}\right)\cos\left(2n\pi\frac{t}{T_0^{(k)}}\right)$$

where $\phi^{(k)}(t)$ is the generic phase variable taking values in the set $\{\alpha_k, \beta_k, \gamma_k\}$, and the sign depends on the mode considered. These results imply that phases remain locked to the initial value for a time interval $\delta\tau^{(k)}$, depending on the mode $k$, and after this time duration abruptly jumps, say $\phi_k(t)$ undergoes a sudden rotation, of a factor $\pm\pi$, thus starting a new locking period, and this behaviour is repeated periodically, due to periodic resetting of phases. As reported in the various panels of Fig 11, when $\mathcal{C}$ increases, all phases oscillate with different periods and different locking duration $\delta\tau$. In particular, in the low-forcing case, e.g. for $\mathcal{C} = 0.23$ (Fig 11, the second column), the various $\phi^{(k)}(t)$ exhibit mixed dynamics where some phases present the step-like locking with different $\delta\tau$, while others remain constantly locked at the initial value for the whole numerical run, moreover such mixed behaviour is also observed for $\gamma_k$. This dynamics persists up to relatively large values of $\mathcal{C}$ (high-forcing regime), where the relative amplitudes are always chaotic, while the time of locking for phases becomes shorter and shorter, as well as the period of oscillation (Fig 11, the third and fourth columns). For these higher values of $\mathcal{C}$, the consecutive phase jumps become very rapid, and the step-like locking loses its periodicity or quasi-periodicity. In addition, the amplitudes recorded in the phase jumps move away from the value $\pm\pi$, as shown in the last two panels of Fig 11, for all $\alpha_k(t)$ (the first row), $\beta_k(t)$ (the second row), and $\gamma_k(t)$ (the third row), where the temporal evolution of the generic $\phi_k(t)$ becomes somewhat chaotic.

To model this complex phase dynamic, a Phase-Transition Curve (PTC) [33, 50–52] can be defined, representing a mapping relation for the generic time instant $t_i$ composing the dynamics of the phase variable $\phi^{(k)}(t_i)$ of the $k$-th mode, namely:

$$\phi_{i+1}^{(k)} = (\phi_i^{(k)} + W_i^{(k)}(\mathcal{C})) \bmod 2\pi \tag{15}$$

evaluated at two different times $\phi_i^{(k)} = \phi^{(k)}(T_i)$ and $\phi_{i+1}^{(k)} = \phi^{(k)}(T_{i+1})$, where $W_i$ is some function describing the phase shift due to the resetting. It is evident that the PTC can play the same role as the Poincaré map in describing the transition to chaos for phase variables.

The PTCs for $\phi_k(t) := \alpha_k(t)$, $\phi_k(t) := \beta_k(t)$, and $\phi_k(t) := \gamma_k(t)$ are shown in the various panels of Fig 12, as a function of different values of the control parameter $\mathcal{C}$, depicting the periodic state for $\mathcal{C} = 0.19$ (Fig 12, the first column), and three different chaotic regimes observed at $\mathcal{C} \in \{0.23, 4.9, 8.0\}$ (Fig 12, the second, third, and fourth columns), respectively. When the system is purely periodic, and no phase shift is present, the PTC is represented as a single fixed point on the bisector of the plane $(\phi(t_i), \phi(t_{i+1}))$, and $W_i^{(k)} = 0$. When the dynamics is step-like locked, or in other words when $\phi_k(t)$ experience the $\pm\pi$ phase-resetting jump, the PTCs appear as an ordered periodic square structure with side length $\pi$, where the phases are locked on two opposite angles on the PTC map (e.g. Fig 12, second row, the first panel). As other $k$ modes begin to develop step-like locking, more and more regular square structures appear on the map (Fig 12, the second column). When amplitudes follow periodic or multi-periodic dynamics, phase variables may experience a narrower or wider distribution of locking time duration

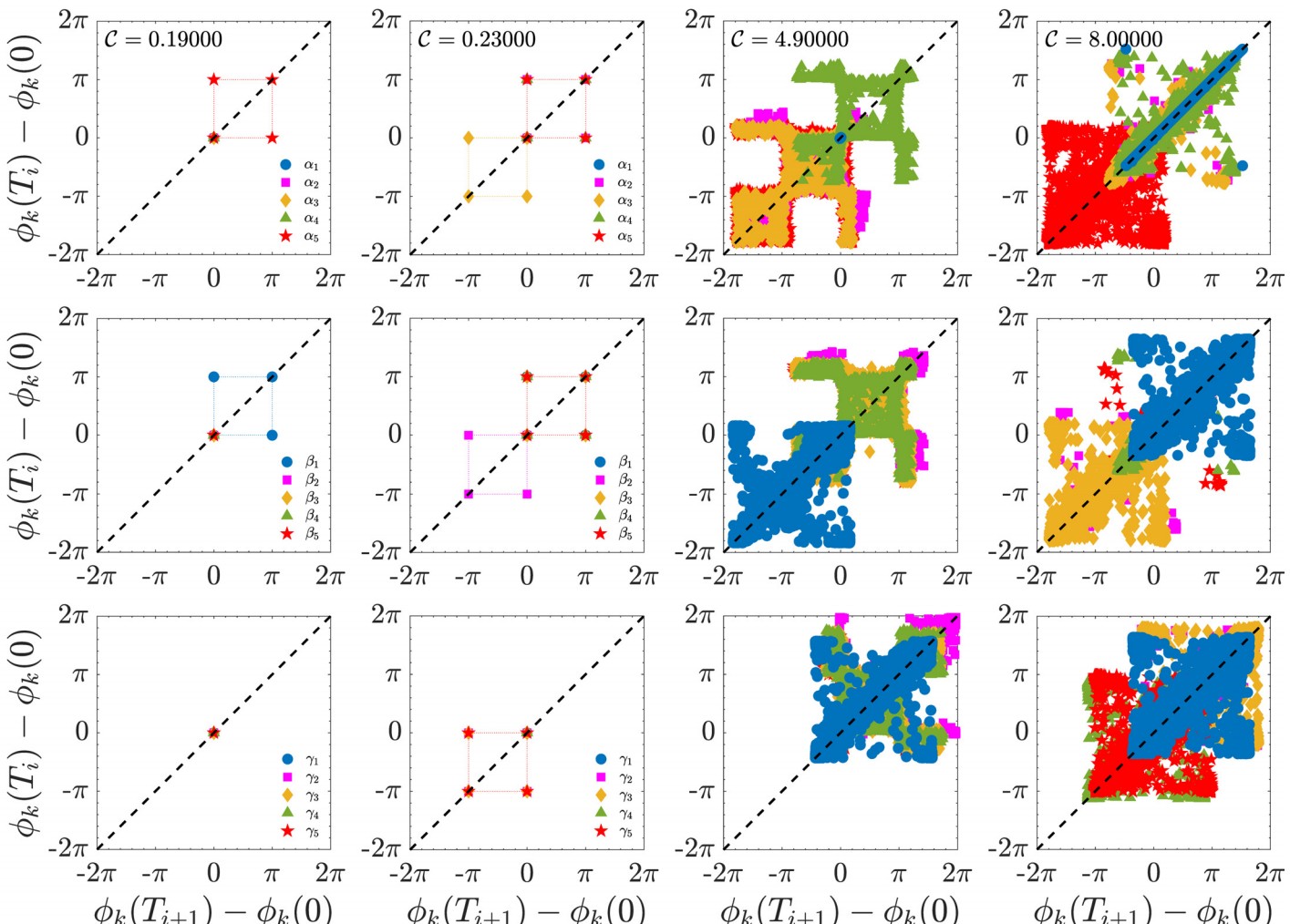

**Fig 12. PTCs for the phase variables $\alpha_k(t)$ (first row), and $\gamma_k(t)$ (second row), as a function of the control parameter $\mathcal{C}$, showing the evolution from low-forcing regime to the higher forcing regime.** The first two columns present the transition from a single-point map (constantly locked dynamic) to a four-point map (step-like locking dynamic). The last two columns report the transition to the chaotic state where the phase shift exceeds the value $\pm\pi$. The dashed line represents the constantly locked dynamic and is reported as a guide for the eye.

$\delta\tau$, respectively, (*e.g.* Fig 12 the third and the fourth column), in such cases, the term $W_i^{(k)} \neq 0$ but rather a more complex behaviour is expected, which can be described via the relation:

$$W_i^{(k)} = s_i^{(k)} \epsilon \, \delta[T_i - (T_0^{(k)} - s_i^{(k)} \Delta T^{(k)})] \qquad (16)$$

being $s_i = (-1)^{i+1}$ for $i = 1, 2, \ldots, \epsilon = \pm\pi$ the amplitude of the step-like jump, and $\delta$ is the traditional Kronecker delta function. It is interesting to note that for $\mathcal{C} = 0.23$, although the amplitude is chaotic, the phase dynamics always remain ordered since this dynamics is constrained to explore the same points on the plane. The phases locking period depends on the external forcing term, the higher $\mathcal{C}$, the smaller the value of $\delta\tau^{(k)}$ is, or in other words, intense control parameter amplitudes generate more frequent jumps.

When the system enters the higher forcing regime, the relation even describes the PTC (16). However, the parameters $\epsilon$, $T_0^{(k)}$ and $\delta\tau^{(k)}$ are not constant anymore, but rather become

time-dependent, or stochastic, depending on $\mathcal{C}$. Interestingly, even for large $\mathcal{C}$, the plane is not fully explored by the system (Fig 12, the third column). Rather, some regions are forbidden for the dynamics, indicating that certain phase-shift resetting is forbidden. For very high values of $\mathcal{C}$, the points of the map are densely distributed around the bisector of the plane, indicating a continuous phase precession interrupted by resets at random times. Since the system does not always reach a $\pm\pi$ shift at these resets, the phases have to precede faster in order to return on the "correct trajectory", thus generating various spurious jumps, described by the scattered points on the plane, whose amplitude can span a range of $\in [0, \pm 2\pi[$ (Fig 12, the third and fourth columns).

Due to the dynamics of single phases described above, the three-phase coupling, which is nothing but a linear combination of phases, follows a similar behaviour. An analogous PTC for the variable $\Phi_{\pm kpq}^{\alpha\beta\gamma}$, can be defined as follows:

$$\Phi_{i+1} = (\Phi_i + Z_i(\mathcal{C})) \bmod 2\pi$$

being $\Phi_i = \Phi_{\pm kpq}^{\alpha\beta\gamma}(T_i)$ the generic triad-phase, in such a way that the phase shift of the PTC function is composed by the sum of three terms, $W_i(k) + W_i^{(p)} + W_i^{(q)}$ for each interaction triad, as in the previous case: Eq (16). In addition, according to (16), every mode has its own phase-shift time $\delta\tau^{(k)}$ and periodicity $T_0^{(k)}$, so that the linear combination of three modes, each with jumps at different times, generate a stochastic dynamics for the three-phase couplings $\Phi_{\pm kpq}^{\alpha\beta\gamma}$ even for shorter values of $\mathcal{C}$.

In Fig 13 is reported the temporal evolution of the cumulative three-phase coupling terms observed to the slowest mode $k = 1$, for the two forcing regimes (low-forcing regime in the first row and high-forcing regime in the second row) for different values of $\mathcal{C}$. The three columns in the figure are relative to the three variables composing the RSW $\mathcal{T}_5$ model: $u$ first column, $v$ central column, and $\eta$ Right column. As can be clearly seen, the effect of the accumulation of continuous phase jumps generates Brownian-like dynamics where the drift seems related to the value of the control parameter $\mathcal{C}$. Such randomization effect becomes more efficient as the value of $\mathcal{C}$ increases since the locking time $\delta\tau$ related to step-like structures decreases. When the system falls on a non-chaotic state, the dynamics of three-phase coupling terms are reduced to a smoother dynamic: constant when the phases are constantly locked or characterized by small drift from an average value when the dynamic is step-like locked or quasi-periodic.

## 6 Discussion and conclusions

As evidenced by the celebrated Lorenz's model [10, 37], the transition to chaos in complex fluid systems is not universal, but each system shows particular peculiarities. In particular, low-order models of fluid dynamics are valuable tools for understanding the route to chaos in turbulent media, as well as large/meso-scale dynamics. Among others, low-order models of triad interactions show proper peculiarities. Despite the fact that complexity depends on the order of the truncation [10, 33, 36–39], such models share some common characteristics, allowing the investigation of specific properties and the behaviour of local and non–local resonant triads.

Here, in particular, a low-order model composed by a Galerkin truncation of the RSW equations in the Fourier space, and taking into account only five modes $\mathcal{T}_5(u, v, \eta)$ has been investigated, in order to understand nonlinear resonant interactions in a shallow-water environment. One of the scope of our work is to realize that there is no need to invoke singularities or noise to explain the complex dynamics observed in the RSW turbulence [53, 54]. The

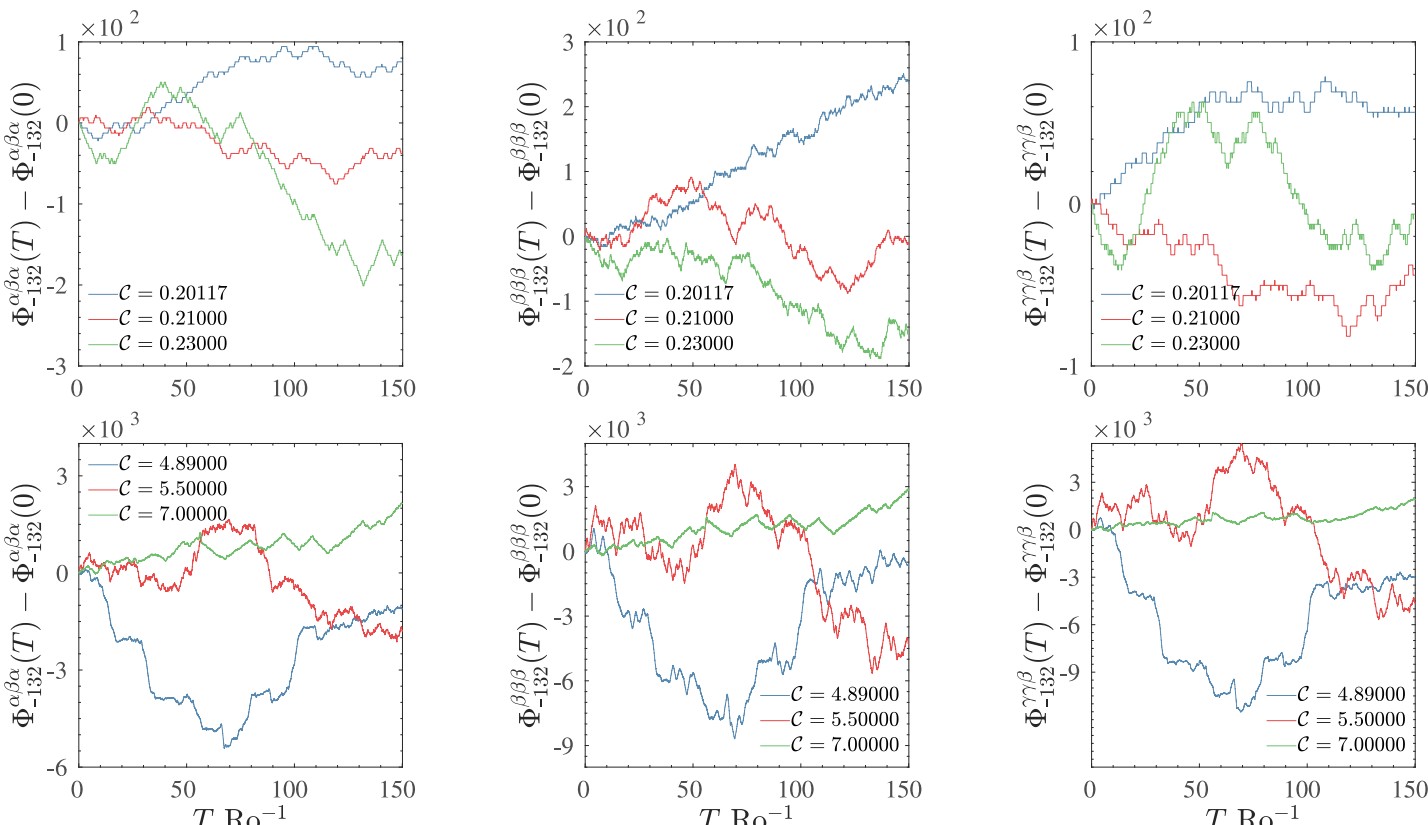

**Fig 13. Temporal dynamic for the cumulative three-phase coupling terms $\Phi_{\pm kpq}^{\alpha\beta\gamma}$, for the mode $k = 1$, in both low–forcing (first row) and high–forcing regime (second row), for different values of $\mathcal{C}$.** The left panels are relative to the $u$-component, central column $v$-component, and right column $\eta$-component. As the control parameter increases, phase jumps become more and more frequent, and the behaviour of $\Phi$ tends to be a Brownian dynamic.

equations contain two kinds of nonlinear interactions related to the usual inertial terms due to both kinetic energy and potential energy. The results are summarized in the following:

- We identified a control parameter $\mathcal{C} = |F_0|\text{Re}$, namely the fraction of Reynolds number indicated by the external forcing term. For small values of the control parameter $\mathcal{C}$, the system presents a transition to chaotic dynamics characterized by a sequence of at least six observable bifurcations. The ratio of the consecutive bifurcation points is equal to $\delta \simeq 4.6$, compatible with the classical Feigenbaum bifurcation cascade process, characterized by $\delta_\infty \approx 4.662\ldots$.

- The physical mechanism of the first transition, in fact, is related to the inertial $(\mathbf{u} \cdot \nabla)\mathbf{u}$ term that dominates the dynamics since the potential energy, in the low–$\mathcal{C}$ regime, is smaller in comparison with the kinetic energy. As the external parameter increases, the system goes back to a regime of quasi-periodic oscillations.

- For higher values of $\mathcal{C}$, a new transition to a chaotic state is observed. At variance with the first one, this second transition is impulsive; in fact, a sharp transition between a quasi-periodic regime and chaos is observed. This secondary chaotic regime is characterized by the appearance of intermittent structures, indicating that the free surface is actively contributing to the turbulent dynamics of the system. In other words, the forcing in the right-hand side of the momentum equation creates waves on $\eta(\mathbf{r}, t)$ that, during their propagation, act in turn as a forcing on the velocity field.

- As far as the dynamics of the phases of the field is concerned, the situation is somewhat peculiar. In fact, by analyzing the phases, for lower values of $\mathcal{C}$, they do not enter the mechanism of chaotic transition. In fact, it has been observed that phases are locked for a certain period of time $\delta\tau$, followed by a sudden transition due to a rotation of $\pm\pi$, and the duration of the locking period $\delta\tau$ depends on $\mathcal{C}$. As $\mathcal{C}$ increases $\delta\tau$ reduces. This dynamic could be due to the presence of an invariant subspace in the Fourier equations, whose dynamics depend only on the amplitudes of the various Fourier coefficients of the fields but not on the phases, namely $\mathcal{T}_5(u, v, \eta) = \mathcal{R}_5(A, B, \Gamma) \cup \mathcal{I}_5(u, v, \eta)$. This subspace is stable, so the phases of the various Fourier modes "prefer" to stay locked to the initial value or to jump by a fixed value in a way that $\mathcal{I}_5(u, v, \eta) = constant$.

- At larger values of $\mathcal{C}$, when the amplitudes undergo the secondary transition to chaos, the invariant subspace seems to be destabilized, and the period of locking of phases becomes very short. In this situation, the phases undergo a continuous precession characterized by random jumps in time, with an amplitude of the order of a fraction of $\pm\pi$. The randomness of time of jumps and amplitude of jumps increases as $\mathcal{C}$ increases. To our knowledge, the differences between the transition to chaos for amplitudes and phases have never been outlined, and this aspect deserves further investigation.

- When a Fourier coefficient of a field is written in terms of amplitude and phase, we can write a set of equations for both quantities. The resulting set of equations contains some triad-phase interactions formed as a sum of phases for each interacting wavevector. Regardless of the values of $\mathcal{C}$, the accumulation of multiple subsequent phase jumps with different uncorrelated periods makes stochastic the dynamics of the triad-phases, even when the amplitudes are periodic. This is a peculiar aspect of phase dynamics that deserves further investigation.

As stated before, it should be remarked again that Galerkin models do not contain, nor shall we be concerned with, the inertial or dissipative ranges of fully developed turbulence, whose description requires an infinite number of wave vectors. The emphasis on the chaotic dynamics of the RSW system presented here should not be confused with fully developed turbulence since the attractors observed in the truncated RSW system should not be confused with the formation of a turbulent cascade so that it is not immediately obvious one finds a counterpart of the chaotic dynamics in the observations of fluid turbulence.

In conclusion, the different transitions to chaos with respect to the injected energy and the coexistence of phase locking and free precession periods are interesting features highlighted by the five-mode RSW model studied here.

Even if further efforts and studies are required to fully understand the detailed nature of RSW turbulence on geostrophic scales, this work opens novel approaches to studying the dynamics and the transition to chaos via multiple bifurcations and intermittent transitions in RSW systems within the general framework of dynamical systems theory.

## Supporting information

**S1 File.**
(ZIP)

## Acknowledgments

We extend our sincere gratitude to the Research Computing department for their invaluable support and cooperation in facilitating our project, `FSU-2023-014`, on the Almesbar HPC cluster under the project code `kuin0099`.

## Author Contributions

**Conceptualization:** Francesco Carbone, Denys Dutykh.

**Data curation:** Francesco Carbone, Denys Dutykh.

**Formal analysis:** Francesco Carbone, Denys Dutykh.

**Funding acquisition:** Francesco Carbone, Denys Dutykh.

**Investigation:** Francesco Carbone, Denys Dutykh.

**Methodology:** Francesco Carbone, Denys Dutykh.

**Project administration:** Francesco Carbone, Denys Dutykh.

**Resources:** Francesco Carbone, Denys Dutykh.

**Software:** Francesco Carbone, Denys Dutykh.

**Supervision:** Francesco Carbone, Denys Dutykh.

**Validation:** Francesco Carbone, Denys Dutykh.

**Visualization:** Francesco Carbone, Denys Dutykh.

**Writing – original draft:** Francesco Carbone, Denys Dutykh.

**Writing – review & editing:** Francesco Carbone, Denys Dutykh.

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
