## [Decision Letter · Decision Letter 0]

22 May 2024

PONE-D-24-14063Route to chaos and resonant triads interaction in a truncated Rotating Nonlinear shallow-water modelPLOS ONE

Dear Dr. Dutykh,

Thank you for submitting your manuscript to PLOS ONE. After careful consideration, we feel that it has merit but does not fully meet PLOS ONE’s publication criteria as it currently stands. Therefore, we invite you to submit a revised version of the manuscript that addresses the points raised during the review process.

We look forward to receiving your revised manuscript.

Kind regards,

Qiang Li

Academic Editor

PLOS ONE

Journal Requirements:

   "FSU-2023-014"

4. In the online submission form, you indicated that "From the Authors under simple request." 

5. Please note that funding information should not appear in any section or other areas of your manuscript. We will only publish funding information present in the Funding Statement section of the online submission form. Please remove any funding-related text from the manuscript.

Reviewers' comments:

Reviewer's Responses to Questions

**Comments to the Author**

1. Is the manuscript technically sound, and do the data support the conclusions?

Reviewer #1: Yes

Reviewer #2: Yes

Reviewer #3: Yes

2. Has the statistical analysis been performed appropriately and rigorously? 

Reviewer #1: Yes

Reviewer #2: Yes

Reviewer #3: Yes

3. Have the authors made all data underlying the findings in their manuscript fully available?

Reviewer #1: Yes

Reviewer #2: Yes

Reviewer #3: Yes

4. Is the manuscript presented in an intelligible fashion and written in standard English?

Reviewer #1: Yes

Reviewer #2: Yes

Reviewer #3: Yes

5. Review Comments to the Author

Reviewer #1: This investigation analyzed two different chaotic behavior transitions through the construction of an autonomous five-mode Galerkin truncated system employing complex variables. A very clear mechanism analysis of chaotic behavior is presented in this investigation, and an excellent time segmented locking behavior also appears in the rotating shallow water model. However, there are still some doubts that need to be answered or a series of content that needs to be added appropriately.

1：All formulas do not indicate the source of references or citations. In addition, the meanings of some letters in the formula are not detailed, which is crucial for understanding the research content.

2：What are the differences between the autonomous five mode Galerkin truncation system constructed in this study and the traditional system for rotating shallow water models. What are the advantages of Galerkin truncation systems？

3：In the abstract section, the trend of “It was observed that the duration of phase stability diminishes with an increase in injected energy, culminating in the onset of chaos within the phase components at high energy levels.” cannot be found in the investigation. Relevant data needs to be listed (line chart).

4: In section 5.2, can phase dynamics be validated by some numerical simulation software to validate the model results?

Reviewer #2: The article presents a rigorous study of chaotic behavior and large-scale phase dynamics in a rotating shallow water model using an autonomous five-model Galerkin truncated system constructed with complex variables. The results point out that there are two distinct transitions to chaotic behavior as the energy level introduced in the system gradually increases. The creation of the first chaotic state is mainly attributed to the dominant role of inertial forces in the nonlinear interactions, while the creation of the second chaotic state is attributed to the enhanced importance of the free water surface elevation in the dynamical process. As described by the authors, the model highlights the interesting features of chaotic transitions with different injected energies and the coexistence of phase-locked and free-forward phases, opening new avenues for the study of chaotic bifurcations in dynamical systems theory and for the understanding of RSW turbulence at the geostrophic scale.

The article is very rich in research with complete modeling and computational analysis. Therefore, I agree to publish it in this journal with only two minor suggestions.

(1) The abstract should state the background of the research.;

(2) Important computational values may appear in the abstract;

(3) The conclusion should be more concise; the current conclusions are somewhat detailed.

Reviewer #3: (1) The abstract of the manuscript needs to be supplemented with appropriate key quantitative results to enhance its persuasiveness. For example, it was mentioned that "It was observed that the duration of phase stability diminishes with an increase in injected energy, culminating in the onset of chaos within the phase components at high energy levels.".

(2) The sentences in lines 4-6 that "Moreover, their interplay is related to essential processes in atmospheric and oceanic sciences, such as transport and exchange of moisture, heat, gaseous tracers or salinity, and momentum, depending on the which of boundary layer being considered." should be supported by citing some works. https://doi.org/10.1016/j.oceaneng.2024.117029, https://doi.org/10.1039/C8RA06430J

(3) The second part of the manuscript presents a description of the model constructed by the research institute. And the statement is very detailed, which is very good. However, how the model was implemented and through what tools. These details also need to be appropriately displayed.

(4) How applicable is the model constructed in the second part to the relevant research content in this manuscript? The reviewer believes that it is necessary for the model to be validated before application.

(5) In the manuscript, it is not friendly to the review process if the author places the main text, figures and tables separately. It is recommended to insert the figures and tables into the corresponding positions of the text when submitting the revised manuscript.

(6) Why is there a significant difference in the variation trend of the curve with T when the C value is high compared to the curve when the C value is low in Figure 13?

6. PLOS authors have the option to publish the peer review history of their article (what does this mean?). If published, this will include your full peer review and any attached files.

Reviewer #1: No

Reviewer #2: No

Reviewer #3: No

---

## [Author Response · Author response to Decision Letter 0]

29 May 2024

See the separate responses in corresponding PDF files.

---

## [Decision Letter · Decision Letter 1]

3 Jun 2024

Route to chaos and resonant triads interaction in a truncated Rotating Nonlinear shallow-water model

PONE-D-24-14063R1

Dear Dr. Dutykh,

We’re pleased to inform you that your manuscript has been judged scientifically suitable for publication and will be formally accepted for publication once it meets all outstanding technical requirements.

Kind regards,

Qiang Li

Academic Editor

PLOS ONE

Additional Editor Comments (optional):

Reviewers' comments:

Reviewer's Responses to Questions

**Comments to the Author**

1. If the authors have adequately addressed your comments raised in a previous round of review and you feel that this manuscript is now acceptable for publication, you may indicate that here to bypass the “Comments to the Author” section, enter your conflict of interest statement in the “Confidential to Editor” section, and submit your "Accept" recommendation.

Reviewer #1: All comments have been addressed

Reviewer #3: All comments have been addressed

2. Is the manuscript technically sound, and do the data support the conclusions?

Reviewer #1: Yes

Reviewer #3: Yes

3. Has the statistical analysis been performed appropriately and rigorously? 

Reviewer #1: Yes

Reviewer #3: Yes

4. Have the authors made all data underlying the findings in their manuscript fully available?

Reviewer #1: Yes

Reviewer #3: Yes

5. Is the manuscript presented in an intelligible fashion and written in standard English?

Reviewer #1: Yes

Reviewer #3: Yes

6. Review Comments to the Author

Reviewer #1: Following the meticulous editing process undertaken by the author, the article has been enhanced in terms of its overall completeness and rigour. It is therefore recommended that it be accepted for publication in Plos One.

Reviewer #3: All my comments have been replied by the authors, and this paper can be accepted for publication now.

7. PLOS authors have the option to publish the peer review history of their article (what does this mean?). If published, this will include your full peer review and any attached files.

Reviewer #1: No

Reviewer #3: No

---

## [Editor Report · Acceptance letter]

14 Jun 2024

PONE-D-24-14063R1 

PLOS ONE

Dear Dr. Dutykh, 

I'm pleased to inform you that your manuscript has been deemed suitable for publication in PLOS ONE. Congratulations! Your manuscript is now being handed over to our production team.

Kind regards, 

on behalf of

Dr. Qiang Li 

Academic Editor

PLOS ONE